# Polling India via regression and post-stratification of non-probability online samples

**Roberto Cerina**[1]*, **Raymond Duch**[2]

**1** Data Analytics and Digitalisation, Maastricht University, Maastricht, Netherlands, **2** Nuffield College, University of Oxford, Oxford, United Kingdom

* r.cerina@maastrichtuniversity.nl

**Data Availability Statement:** The data and replication code have been deposited on the Harvard Dataverse replication site. https://doi.org/10.7910/DVN/0RRVKJ.

**Funding:** RC and RD received funding from Nuffield College from the Research Allowance of

## Abstract

Recent technological advances have facilitated the collection of large-scale administrative data and the online surveying of the Indian population. Building on these we propose a strategy for more robust, frequent and transparent projections of the Indian vote during the campaign. We execute a modified MrP model of Indian vote preferences that proposes innovations to each of its three core components: stratification frame, training data, and a learner. For the post-stratification frame we propose a novel Data Integration approach that allows the simultaneous estimation of counts from multiple complementary sources, such as census tables and auxiliary surveys. For the training data we assemble panels of respondents from two unorthodox online populations: Amazon Mechanical Turks workers and Facebook users. And as a modeling tool, we replace the Bayesian multilevel regression learner with Random Forests. Our 2019 pre-election forecasts for the two largest Lok Sahba coalitions were very close to actual outcomes: we predicted 41.8% for the NDA, against an observed value of 45.0% and 30.8% for the UPA against an observed vote share of just under 31.3%. Our uniform-swing seat projection outperforms other pollsters—we had the lowest absolute error of 89 seats (along with a poll from 'Jan Ki Baat'); the lowest error on the NDA-UPA lead (a mere 8 seats), and we are the only pollster that can capture real-time preference shifts due to salient campaign events.

## 1 Introduction

Much of social science research involves constructing samples and implementing statistical estimators that allow us to use these samples to say something about the population. Public-opinion estimation in the run-up of an election is a case in point; in fact, it accounts for many of the recent innovations in sampling and prediction technologies. We contend that these innovations will have an important effect on estimating public opinion in contexts where sampling and data collection are particularly challenging. India is one of these challenging contexts. This essay reports our forecasts of the 2019 India Lok Sabha elections. Our

RD. The funder URL is https://www.ox.ac.uk/admissions/graduate/colleges/nuffield-college. The funder had no role in study design, data. collection and analysis, decision to publish, or preparation of the manuscript.

**Competing interests:** The authors have declared that no competing interests exist.

methodological contribution develops innovations in data collection, data enhancements, and modeling to address the challenges of election forecasting in contexts such as India.

We propose a sound, expedient and accessible guide for opinion researchers to analyze pre-election attitudes in countries like India, employing non-representative samples of voters. We apply a variant of Multilevel Regression and Post-stratification (MrP) to Indian convenience samples gathered in the run-up of the 2019 Lok Sabha election. The initial and classic contribution of MrP "technology" is the generation of small-area estimation from nationally representative surveys [1–3]. Typically, a nationally representative sample of standard size (e.g. 1,000 to 2,000 units) is obtained via traditional sampling methods (e.g. RDD). The small area estimates that are then obtained via MrP are typically preferable to disaggregated estimates from the sample [1]. Our efforts will concentrate on producing a national-level forecast, rather than small area estimation; we employ MrP-like estimation to increase the robustness of our national-level predictions.

The first non-standard MrP challenge we face in India is a convenience sample that is decidedly non-representative. There are applications of MrP methods that address this challenge and we build on them in this essay. The example with the most extreme unrepresentative sample to date is [4]: a sample of Xbox gamers. In the case of extremely unrepresentative samples, the role played by MrP is to account for observable selection effects by including variables which are correlated with propensity to respond and the outcome of interest as predictors in the regression model. The Xbox study successfully addressed this estimation challenge by collecting close to 800,000 responses from around 400,000 gamers [4]. Similarly, [5] YouGov's studies (2016—EU Ref; 2016—US election; 2017 UK election; 2019 UK election) leveraged online panels as large as 100,000 respondents. Very large-N convenience samples paired with MrP estimation can successfully account for observed selection effects.

Our Indian election forecast application will rely on a decidedly unrepresentative convenience sample; but even more challenging is the sample's relatively modest size. We will show that even small-N very unrepresentative samples can be used to obtain estimates of national vote intentions that are well within acceptable deviations from actual vote outcomes.

MrP estimation presumes a comprehensive stratification frame. A second, novel MrP challenge that we faced in India is incomplete and poor-quality data for constructing our stratification frame. Without complete and accurate information on the joint distributions of all relevant covarites, we need to create synthetic stratification frames with the data available [3]. We provide a strategy for creating a robust stratification frame by combining multiple complementary data sources.

Hence, we will use variations on classic MrP estimation strategies to generate national-level vote-share predictions for the Indian 2019 Lok Sabha election, during the duration of the campaign. There are three key elements to the execution of our MrP modeling: a) a stratification frame: a vector of counts for a set of cells, or mutually-exclusive and exhaustive voter categories, in the population of interest; b) training data: an individual-level dataset that allows for demographic profiles to be correlated with outcome-behaviour of interest—in our case, voting behaviour; c) a learner: an algorithm which can best summarize the relationships implied by the individual- level dataset and coherently project these onto voter-categories we have not yet seen in the sample—also known as 'out-of-sample' predictions. This paper contributes to the literature on each of these key elements, by innovating the frames, training-data and learners needed to generate accurate readings of public opinion.

We propose a novel treatment of stratification frames, motivated by the challenges faced in the Indian case-study. While in traditional settings such as the UK or the US we would rely on census micro-data—random 1% samples of the decennial census—to generate deep and precise frames, we cannot do so in India, as the micro-data is not readily available. To optimize

the depth—precision trade-off in our Indian stratification frame, we propose a novel Data Integration approach [6], that allows the simultaneous estimation of counts from multiple complementary sources, such as census tables and auxiliary surveys. This innovation is similar in implementation, though motivated by substantially different reasoning, to Cerina and Duch [7], and contributes to the literature dealing with the augmentation of stratification frames [3].

The training data proposed here are also innovative—we assemble panels of respondents from two unorthodox online populations: Amazon Mechanical Turks workers and Facebook users. We contribute to the literature on online non-probability sampling in India, by showing persistent bias in geographic distribution, gender, education and income, as well as evidence for differential online selection effects based on sampling medium.

Finally, we replace the Bayesian multilevel regression learner with Random Forests [8] (RFs). This is motivated by three considerations: i) Our goal is to facilitate real-time updating of our vote forecasts. Random forests can grow trees in parallel and deploy memory-optimized tree-growing algorithms which facilitate daily fitting and de-bugging of our predictions [9]. ii) We sought to address the model specification challenges associated with MrP. Random Forest estimators can optimize these modeling decisions, especially when it comes to choosing amongst a large number of area-predictors [10] that are interacted with individual-level variables [11]. iii) An important aim is to shrink prediction variance recognizing that in doing so we pay a price in terms of bias. By implementing a Random Forest bagging estimator we could apply regularization to our predictions [12].

As a result of incorporating these innovations, our pre-election forecasts of the 2019 Lok Sahba elections were highly robust. The predicted point estimate for the NDA is 41.8%, against an observed value of 45.0% (prediction error of 3.2% points) and the point estimate prediction for the UPA is just under 30.8% against an observed vote share of just under 31.3% (prediction error of around 0.5%). Our uniform-swing projection outperforms other pollsters: our prediction has the lowest absolute error, 89 seats (only one other forecast approximates this performance), and the lowest error on the NDA-UPA lead, a mere 8 seats, which is the pivotal quantity to understand who governs when the dust settles. Our UPA seat share forecast is only off by 18 seats and off by 26 for the NDA.

The article is organized as follows: Section 2 sketches a picture of the 2019 election, outlining some of the general challenges which make this an important, albeit difficult, exercise in psephology; Section 3 presents the various data sources we use at each step of our estimation procedure for custom built stratification frames; we outline a systematic procedure to obtain reliable cell-level counts for the frame. Section 4 proposes our novel approach to collecting and using convenience samples for training data. Section 5 outlines models to estimate cell-level turnout and voting behavior. The results of our predictions are reported in Section 6. Finally, we provide a discussion of major contributions, limitations and outline areas of future work in Section 7.

This study involving Human Subjects was approved by the Research Ethics Committee of the Department of Sociology University of Oxford. The approval number is Ref. `SOC_R2_001_C1A_19_07`. We received informed consent online from subjects. They were provided with the consent form at the outset of the survey—they were then asked to confirm that they had read the consent form; and then that they agreed to the terms of the consent document. Unless they answered yes they were not permitted to participate in the study.

## 2 Forecasting the 2019 Lok Sabha

Voting behaviour in India has long been the subject of academic scrutiny: Eldersveld [13] first exported the 'National Election Study' model, pioneered in the US [14], to the world's largest

democracy; Butler et al. [15] analyzed the patterns of swing over the first 40 years of Indian democracy, introducing a framework of analysis that persists in current scholarship [16]. Despite this substantial tradition in the study of electoral behaviour, Indian pre-election opinion polls have not performed particularly well in predicting electoral outcomes [17, 18]. This is hardly surprising given the sheer size and heterogeneity of the Indian electorate that can reduce statistical power at the sub-national level but also makes it very challenging to sample these diverse and disparate socio-demographic groups. More recently, the advent of the internet and advanced technological tools has enabled the collection of large-scale, historical administrative data and also facilitated large-scale online surveying of the population. These advances have opened the door to more fine-grained and innovative analyses of the Indian vote [19].

In spite of these advances, forecasting Indian elections remains challenging. Two major alliances faced off in the 2019 election in India: the National Democratic Alliance (NDA) led by incumbent Prime Minister Narendra Modi, and the United Progressive Alliance (UPA) led by opposition leader Rahul Gandhi. The country elects members of parliament via a first-past-the-post system, and the coalition which has a majority of the 543 contested seats in the Lok Sabha usually nominates a prime minister. A trivial number seats are assigned by the president. In 2014 the NDA captured a large and unforeseen victory, winning 336 seats, whilst the UPA heavily under-performed, only winning 59 seats. Polling error on the NDA result was enormous, averaging at an under-estimation of around 88 seats for the last three published polls, which happened also to be the most accurate (in relative terms). Our methods vastly improve on this historical performance.

We propose an innovative regression and post-stratification modeling strategy to address the challenges of election forecasting in contexts such as India. Our goal is to predict accurately the 2019 Lok Sabha vote choice, $\hat{v}$, of the Indian voters. Eq (1) summarizes the quantities we estimate in order to obtain this vote prediction. The last term in the numerator, $\hat{N}_i(X)$ represents elements from a constructed stratification frame, typically obtained from a national census micro-data file. The first two elements of the numerator, $\sum_i \hat{Pr}_i(V \mid T, X) \times \hat{Pr}_i(T \mid X)$ are vote choice and turnout probabilities, typically estimated from public opinion polls:

$$\hat{v} = \frac{\sum_i \hat{Pr}_i(V \mid T, X) \times \hat{Pr}_i(T \mid X) \times \hat{N}_i(X)}{\sum_i \hat{Pr}_i(T \mid X) \times \hat{N}_i(X)}. \tag{1}$$

Note that to simplify the initial motivation here we have omitted some notation to avoid repeating redundant indices. The complete notation that we will elaborate on below is the following: $\hat{v}$ is a vector of length $J = 3$, bounded in [0, 1] indicating the vote-share of the three main alliances of interest; $V$ is a multivariate, dichotomous random variable, indicating the event that a given population-cell $i$ votes for alliance $j$—hence taking value $V_{ij} = 1$ if cell $i$ votes for party $j$, and $V_{ij} = 0$ otherwise. $T$ is a dichotomous random variable indicating the event that cell $i$ turns out to vote, whilst $X$ is the design matrix describing each cell $i$ according to a set of covariates, such that a cell $i$ is uniquely identifiable by a combination of values from $X$.

But India is a challenging context for generating vote forecasting based on regression and post-stratification because of data issues. The last, stratification frame, term in the numerator of Eq (1) should be based on a recent detailed enumeration of the population—typically a census micro-file. This is the case with most applications in the regression and post-stratification literature (see [1, 20–22], to cite a few). In the India case, we cannot rely on off-the-shelf stratification frames derived from the census. We have developed a strategy to piece together a new and exhaustive frame from multiple sources of imperfect data on the Indian population. Table 1 summarizes the data sources we rely on for our forecasts. Our "synthetic" stratification

**Table 1. A summary table of data sources by some meta-characteristics, including: Dates of collection; representative quality; depth; sample size; whether they include demographic or political variables; whether they are leveraged to build the stratification frame and/or the voting behaviour models.** Note that though the census tables enumerate the entire population of the country at the macro-level, we sample at random from its joint distribution a smaller micro-data sample of size noted in the table, which we use later to integrate the IHDS.

|  | IHDS | Census | INES | Lok Dhaba | Facebook Users | Mechanical Turks |
|---|---|---|---|---|---|---|
| Dates | Nov. 2011—Oct. 2012 | Apr. 2010—Mar. 2011 | Apr.—May 2014 | Historic | Feb.—Apr. 2019 | Feb.—Apr. 2019 |
| Sampling | Probability | Probability | Non-Probability | Electorate | Non-Probability | Non-Probability |
| Depth | Micro | Micro | Micro | Macro | Micro | Micro |
| N | 135, 984 | 3, 800, 213 | 22, 295 | Electorate | 1, 633 | 5, 807 |
| Demographic | ✓ | ✓ | ✓ |  | ✓ | ✓ |
| Political |  |  | ✓ | ✓ | ✓ | ✓ |
| Strat. Frame | ✓ | ✓ | ✓ | ✓ |  |  |
| T. Model |  |  | ✓ | ✓ |  |  |
| V. Model |  |  |  | ✓ | ✓ | ✓ |

frame is derived from four data sources: the IHDS, the Census, the Indian National Election Study and aggregate results for Lok Sahba elections.

The vote choice and turnout probabilities in Eq (1), $(\sum_i \hat{\Pr}_i(V \mid T, X) \times \hat{\Pr}_i(T \mid X))$, are typically estimated with national probability survey samples (although see [7]). In India, it is logistically and statistically difficult to obtain representative samples of voters: surveys have to be designed in multiple languages; rural voters may only be reached by face-to-face interviews due to lack of communications technology; alliances involve local parties and hence in-depth knowledge of local politics is necessary. There are daunting challenges that need to be addressed: within each state, voters speak some subset of the 23 officially recognised languages, and uncountable number of unofficial dialects. According to the World Bank, around 65% of the population lives in rural areas (https://data.worldbank.org/indicator/SP.RUR.TOTL.ZS?locations=IN). To cope with the logistics associated with ensuring each member of this large and diverse electorate has the opportunity to cast their ballot, the election is usually ran over multiple weeks, each constituency assigned to a *phase*, meaning different parts of the countries vote at different times, and campaign effects can sway public opinion during the voting period. The 2019 election was scheduled to be completed in 7 phases, spanning over 5 weeks.

As an alternative we propose a non-probability sampling strategy that relies on online respondents.

Hence, the India vote probabilities in Eq (1) are based on online convenience samples. As Table 1 indicates, we rely on multiple modes of collection, namely searching for subjects amongst Facebook users and Amazon Mechanical Turk workers. There is evidence that relying on multiple loci for data collection can lead to more robust estimates of effects of interest [23], due to the differential selection effects canceling out over loci; furthermore, we rely on the well established literature in aggregating election forecasts [24, 25], and expect that averaging loci-specific selection effects will improve the mean-squared predictive error of our predictions. The turnout probabilities are estimated from the Indian National Election Study combined with aggregate Lok Sahba election statistics.

## 3 Constructing a stratification frame

A stratification frame is a vector of counts, where each element represents a *cell*. Cells are defined as mutually exclusive sets of individual characteristics which uniquely identify a *type* of interest; examples of individual characteristics include gender, age, religious affiliation,

caste status, educational attainment, household income etc. Hence an example of the information contained in a stratification frame would be that there are 1, 209 Hindu females from a backwards caste of age between 35 and 45, with no formal education, living in a household which makes less than Rs 60, 000 per year. Stratification frames have depth and precision.

The depth of the cells is determined by how many different independent variables are necessary to model the outcome of interest—so in our implementation, vote choice. There are two key considerations here: first, as the heterogeneity of the outcome variable in the population increases, we require additional independent variables in order to distinguish cross-cell behaviour in the stratification frame. So in our case, if differences in vote choice are entirely explained by education then we can have a very shallow stratification frame. A second important consideration here is the size and diversity of the survey data used to model the outcome variable. Larger survey sample sizes, typically, increase the number of independent variables that we can include in our modeling of the outcome variable—statistical power will increase with the sample size. The depth of the stratification frame will be directly related to the number of independent variables in these survey-based models predicting the outcome variable. If the survey has a large number of observations, allowing for a rich set of covariates in the modeling of outcomes, then the stratification frame will need to be deep.

Stratification frames contain counts that enable researchers to generate unbiased estimates of the true probabilities of inclusion of each cell in the population at large. We typically think of these stratification frames as being precise; estimating cell-counts with low uncertainty. Highly precise frames can usually be obtained via a country's census. Access to the complete census micro-file would provide a highly precise tally of the number of individuals included in any cell, subject of course to the demographic profiles collected by the census authority. In data-rich environments such as the United States or the United Kingdom, census micro-data, usually around 1% of the anonymized individual-level census responses, are available for researchers to use, enabling the construction of deep and precise frames.

A more likely case is a stratification with cell counts that are estimated with considerable uncertainty. In many cases only cross-tabulations of the data are available. A drawback of this source is that it is usually limited in its depth. For example, the most extensive Indian census cross-tabulations—available at http://www.censusindia.gov.in—provide reasonable depth for geographic distribution and its interaction with age, gender and educational attainment. Other subsets of these demographic characteristics are available, but none that we consider exhaustive. The cell structure of stratification frames is defined by the outcome that is being estimated; in our case vote choice for the 2019 Lok Sabha election. Forecasting vote choice imposes specific demands on the stratification design.

The depth of the stratification frame should be determined by the modeling strategy—in our Indian case, variables that predict vote preference. Our forecast concentrates on national vote shares and headline seat numbers. But it is well-known that Indian political behavior varies significantly by state [16, 26]. Hence, even though we are not producing an area level forecast, we include state-level variables in our cell-definition. As a result, our frame should be accurate at the state-level.

Recall that the cells of the stratification frame should replicate the individual-level independent variables in our 2019 Indian election vote choice model. Political variables, such as past turnout and previous election vote-choice, are known to be highly predictive of current political behaviour in typical applications of model-based pre-election opinion polling [5]. Accordingly, turnout and vote choice from previous elections should be included in the design of the stratification frame. Most census-based micro data do not include these variables which poses a challenge for regression and post-stratification election forecasting. The Indian case is, not surprisingly, no exception.

The cell-level forecasts in our stratification frame are generated by modeling individual-level survey data. Again, the depth of the cell levels in our stratification frame should reflect the model specification for the individual-level survey data. There is a trade-off of course here since precision of this estimation will be constrained by the amount, and depth, of the training data, i.e., the individual-level survey data. At some point, of course, as the cell-sizes decrease, there will not be sufficient statistical power to capture relevant marginal effects. We deploy bagged regression trees [12] models in Section 5 which can perform automatic variable selection and heavy regularization, hence ensuring prediction noise levels are controlled.

Our final national forecast is a weighted aggregation of cell-level forecasts. We build on the literature dealing with forecasting of aggregate economic indicators from disaggregated measures [27–32]. Hence, our prior is that an aggregate-level forecast obtained by combining multiple disaggregate forecasts will be at least as accurate as one generated solely with aggregate-level variables.

The precision of these aggregated cell frequencies of course is determined by the accuracy of our individual cell counts in the stratification frame. In many cases, and our Indian forecast is one of them, the data sources used to populate these cells are incomplete and dated. In our Indian forecast, we only have access to dated census cross-tabulations on a limited number of variables. In order to enhance the precision of the stratification frame cell counts we recommend data augmentation strategies that draw on diverse large scale population surveys. In some cases, very large commercial and public administrative databases can be invaluable for constructing stratification frames. An excellent illustration from the U.S. case is [33]. The large representative surveys and census data identified in Table 1 are used to construct our Indian stratification frame. We rely heavily on the second wave of the India Human Development Survey [34] (IHDS) which is a nationally representative survey of 135, 986 voting-age individuals from 42, 152 households, conducted between November 2011 and October 2012. The IHDS has limitations: It does not include any political variables that are critical to these election forecasts. Second, the IHDS is not representative at the area-level. This is an issue because we have emphasized the importance of the stratification frame being accurate at the deep-cell level in order to generate accurate aggregate level forecasts.

The challenge then is to improve the representative quality of the IHDS at the deep-cell level, specifically at the state-level; and, secondly, augment its covariate-space to include political variables. To improve area-level cell density, [5] propose to use Iterative Proportional Fitting (IPF) [35], also known as 'raking', to re-weight the data according to known area-level frequencies, whilst preserving the underlying interactive structure implied by the original micro-data. Leemann et al. [3] propose to augment the covariate-space of the frame by using nationally representative auxiliary surveys containing overlapping covariates with the existing stratification frame, as well as augmentation-variables of interest. They achieve augmentation by re-weighting the auxiliary survey to match the known area-level distribution of the overlapping covariates. They then simply record the resulting adjusted distribution of the augmentation variables, for each area of interest.

Building on these methods, we propose a novel methodology that leverages data integration via multiple imputation [6]. We integrate the four data sets highlighted in Table 1: IHDS, Census, INES and Lok Dbaha. We now describe the three elements of the integration method: stacking, multiple imputation and iterative proportional fitting.

## 3.1 Stacking

The IHDS micro-data are the point of departure for building our stratification frame. We generate cell counts for cells in the IHDS that are defined by the joint distribution of *{State—Zone*

*—Gender—Age—Educational Attainment—Family Role—Marital Status—Rurality—Religion —Literacy—Caste—Yearly Household Income}.* Note that these cells are deeper than necessary for the vote choice model but we select a surplus of variables to help with multiple-imputation.

The IHDS dataset has limitations. First, the marginal distribution of each variable by state is unrepresentative of the true state population. A second related limitation is that the underlying interactive structure across the variables, i.e. the way in which educational attainment is correlated with caste, will be spurious at the state-level. A third concern is that the IHDS does not include political variables such as past vote-choice and turnout behaviour. We address all three of these issues by implementing a large data-integration technique: we merge the IHDS with data from the decennial census; the Indian National Election Study (INES) [36] and the Lok Dhaba (https://tcpd.ashoka.edu.in/projects/) [19] archive of past election results.

First, we update the underlying interactive structure of the IHDS variables. We want the underlying joint distribution at the state-level to match that of the population. Our state population targets for the joint distribution are derived from the deepest decennial census tables we can find, namely tables containing counts for cells defined as *{State—Zone—Gender—Age— Educational Attainment}*. Tables containing caste, arguably the most predictive variable for vote-choice a-priori, were not as deep, failing to contain a breakdown by age and gender; moreover, the caste classification used in the census was different from that used in the IHDS, and hence was not immediately reconcileable. We could in principle directly rake the IHDS to these census targets, though a problem emerges—namely some of the voter-categories that exist in the census do not exist in the IHDS. This is due to some of these cells being relatively rare in the population, and hence not featuring in the relatively small random sample collected by the IHDS. To avoid instability in raking weights [37], we choose to populate these missing cells using a stacking procedure.

The stacking procedure begins by creating a "census" micro-data set from the joint-distribution tables provided by the Indian census agency. We know each cell's joint-distribution (defined by state, zone, gender, age and education) in the population. This determines this cell's probability of selection into a random sample of the population. We extract a 0.5% random-sample of this population which gives us a micro-data sample of 3, 800, 213 individuals. This new sample size is defined in order to be large-enough to include the largest cells from every state.

A second step in this procedure stacks this "census" micro-data sample below the full IHDS data, treating variables that are present in the IHDS but not in the census sample as 'missing' (this will be the case, for example, for the religion or caste variables). This leads to a new augmented dataset of size 3, 936, 197, with a swath of missing data for records coming from the census. The resulting joint-distribution for the overlapping variables is now dominated by the census and reflects their actual distribution in the population. The price we pay for correcting the joint distribution is that we now have a large amount of missing values; we deal with this problem at the end of this Section.

Our final step in the stacking procedure augments the covariate space with political variables, and, in particular, turnout and vote-choice from 2014. As Table 1 indicates, we use the post-election National Election Study (NES) from 2014 for this augmentation exercize. It contains cells defined by the joint distribution of *{State—Zone—Gender—Age—Religion—Caste— Educational Attainment—Yearly Household Income—2014 Turnout—2014 Alliance Vote}*. We perform the same stacking procedure, treating non-overlapping variables as entirely missing. Once we append the NES to the augmented dataset from the previous two steps we have a complete augmented data set with 3, 958, 492 records; although with a significant number of completely missing values. S2 Table in S1 Appendix shows a subset of the stacked dataset.

## 3.2 Multiple imputation

We propose an imputation strategy for estimating quantities for the large number of missing values in this augmented dataset. To do so, we rely on the Multiple Imputation via Chained Equations (MICE) [38] paradigm. [6], identify two pre-requisites for the MICE procedure to produce meaningful imputed values for the stacked object. First, the values should be 'missing at random' [39]. This is a plausible characterization of the three datasets we used for constructing the augmented dataset. Second, these datasets should share a data generating process. We assume this is true a-priori given that all three datasets are designed to be representative of the true population. The INES stands out as a possible exception to this common DGP assumption. While the dataset is largely representative of demographics, it is almost exclusively a sample of likely voters, with 91% of respondents reporting to have cast a ballot in 2014, highlighting important selection-effects. To overcome this problem, we rake the INES data to the known 2014 election results, at the state level, hence recovering the correct vote-choice and turnout marginals at the state-level. Raking works by re-weighting the observations in the INES to match the marginal distribution of each of our target variables (turnout and vote-choice over the three alliances) in each state. The target distributions are derived by the official 2014 election results in each state; we then proceed to take a bootstrap sample of the dataset, using the standardized raking weights as the sampling probabilities. The resulting dataset can then reasonably be thought of sharing a DGP with the IHDS and the census, though deeper selection issues relating to the INES are to be expected. Deeper selection issues here refer to non-observable selection effects, which cannot be mitigated by simple re-weighting. We have reason to believe that, even conditional on all observable attributes, respondents of the INES will be different from comparable profiles in the population. We suspect this is the case based on the recent experiences with opinion-polling in the UK and US, where we find selection into non-representative samples (of which the INES clearly is one) is correlated with typically non-measured attributes such as social trust [40]. We consider large biases in turnout to be an important indicator of potential selection on non-observables.

MICE can be extraordinarily computationally expensive, as it seeks to model a series of full-conditional distributions over multiple iterations. `missForest` [41] is a MICE-based algorithm designed to deal with high-throughput data, and hence more ameanable to processing our large-dataset. `missForest` uses Random Forests [8] to predict missing values at each MICE step. It is preferable to parametric MICE as it performs automatic variable selection and regularization; can deal with mixed-data types; and has been shown to out-perform traditional imputation models under multiple settings [42]. To further improve the speed of our imputation model, we rely on the `ranger` [9] package for the `R` [43] statistical programming language to train the random forests at each MICE step; the `ranger` algorithm leverages a `C++` backbone to optimize memory-handling and processing of Random Forests, enabling a fast implementation in high-dimensional settings. The `missRanger` [44] package enables us to implement `missForest` with the superior `ranger` algorithm. We can control a number of hyper-parameters in `missRanger`; we limit ourselves to increasing the number of trees to 1, 000 (`num.trees = 1000`), a number we consider large-enough to preclude Monte-Carlo effects in the imputation estimates, while keeping other relevant parameters as the `ranger` default; each tree is fit to a complete bootstrap-sample of the original data (`sample.fraction = 1`, `replace = TRUE`); no maximal depth is imposed on each tree (`max.depth = NULL`); and all available (complete and incomplete) covariates are used as split-candidates.

To make the imputation more robust, we introduce a number of state-level predictors, including: past election results at the state-level from the Lok Dhaba archive; demographic

characteristics derived from the census; and a spatial matrix indicating each state's distance from any other state—in order to account for a degree of local-smoothing [45]. The total number of area-level predictor variables is 175: 35 are distances from other states; 32 are past-vote variables, including, for example, past vote by alliance at the state and zone level, as well as past turnout, for the last three elections; and the rest are zone- and state-level aggregation of individual-level demographics. The algorithm takes approximately 12 hours to converge after 6 iterations, on a `r4.4xlarge` AWS EC2 instance—details on Instance types are available on the AWS website: https://aws.amazon.com/ec2/instance-types/.

### 3.3 Iterative proportional fitting

Having performed multiple imputation, we now have a complete dataset with 3, 958, 492 individual records. But this constructed sample of the Indian population remains unrepresentative at the state-level. We relied on information from the IHDS and INES to impute much of the state-level joint-distributions. And as we pointed out the INES estimates of vote choice and turnout have limitations; both the IHDS and the INES lacked statistical power for many of the joint-distributions estimated; and of course the MICE estimations are likely to be quite noisy given the volume of missing values in the data. S1 and S2 Figs in S1 Appendix show the distribution of 2014 voting behaviour at the zone and state levels described by the post-imputation stratification frame. The Indian states are organized into six zones: Northern, North Eastern, Central, Eastern, Western and Southern. The state-level imputation is poor. At the zone-level the imputations perform better; errors ranging between 7 and 9 points on average for the two main parties. At the zone-level the imputed turnout is reasonably accurate, though clearly the imputed turnout is larger than the true 2014 turnout across the board, revealing perhaps further evidence of INES selection effects and over-reporting. It is worth noting that we do not need these imputations to be perfect at the area-level because we will aggregate over these area-level estimates to produce a national-level forecast. Nevertheless, we want our frame to be as representative of the true population as possible at the deep-cell level in order to allow for robust disaggregated forecasts.

To improve the representation of the dataset at the deep-cell level we employ IPF, as suggested by Lauderdale et al. [5]. The IPF iteratively re-weights each individual in our large post-imputation micro-frame, until these conform to a target set of distributions. The population targets are for our most precisely measured variables: a) the joint distribution of state, age, gender and education as reported in the census tables; and b) the joint distribution of state and 2014 voting-behaviour. IPF is implemented with the R package `anesrake` [46]. We rake areas independently, in order to speed-up convergence and avoid having the algorithm reconcile distributions at different levels. As a result, raking weights could change the relative sizes of the states. To ensure the states remain consistently sized, we re-scale the weights to match the size of each state implied by the census.

S3 and S4 Figs in S1 Appendix show the resulting match between the observed 2014 voting behaviour and our post-imputation and post-raking frame, at the zone and state-level respectively. Typically, state-level 2014 voting behaviour in the frame now exactly matches actual voting outcomes. We do less well matching state-level turnout in the frame compared to actual turnout statistics; though vastly better than the pre-raking estimates. The poor state-level turnout performance is largely a result of the limited turnout training data we have. Recall, the INES had little reported non-voting. Our efforts to correct for this by taking a re-weighted bootstrap sample were not entirely successful. As a result, non-voters are under-represented in the post-imputation frame. This limits the re-weighting options of the IPF algorithm,

**Table 2. Subset of the complete stratification frame, aggregated at the cell-level.**

| Weight | States | Zones | Gender | Age_cat | Religion | Education_level | Caste | Vote_14 | ⋯ |
|---|---|---|---|---|---|---|---|---|---|
| 131.92 | Jammu & Kashmir | North | Male | (54–64] | Muslim | No Formal Edu | Other Backward Castes (OBC) | UPA | ⋯ |
| 175.26 | Jammu & Kashmir | North | Female | (44–54] | Muslim | No Formal Edu | Other Backward Castes (OBC) | UPA | ⋯ |
| 345.93 | Jammu & Kashmir | North | Male | (17–24] | Muslim | Middle or Secondary | Other Backward Castes (OBC) | OTHER | ⋯ |
| 982.97 | Jammu & Kashmir | North | Male | (24–34] | Muslim | Middle or Secondary | Other Backward Castes (OBC) | NDA | ⋯ |
| 127.06 | Jammu & Kashmir | North | Female | (24–34] | Muslim | Middle or Secondary | Other Backward Castes (OBC) | UPA | ⋯ |
| ⋮ | ⋮ | ⋮ | ⋮ | ⋮ | ⋮ | ⋮ | ⋮ | ⋮ | ⋮ |
| 48.04 | Anadman/Nicobar | Eastern | Female | (54–64] | Hindu | Middle or Secondary | Forward/General (except Brahmin) | OTHER | ⋯ |
| 161.28 | Anadman/Nicobar | Eastern | Male | (34–44] | Hindu | No Formal Edu | Forward/General (except Brahmin) | NDA | ⋯ |
| 125.88 | Anadman/Nicobar | Eastern | Male | (24–34] | Hindu | Some Graduate or Higher | Forward/General (except Brahmin) | NDA | ⋯ |
| 35.73 | Anadman/Nicobar | Eastern | Male | (24–34] | Hindu | Middle or Secondary | Forward/General (except Brahmin) | OTHER | ⋯ |
| 40.57 | Anadman/Nicobar | Eastern | Male | (24–34] | Hindu | Primary or Lower | Other Backward Castes (OBC) | OTHER | ⋯ |

effectively precluding a perfect match. Nevertheless, the match is perfect at the zone-level, and close-to-perfect at the state-level for the major parties.

As a final step in the construction of a representative stratification frame for the 2019 Lok Sabha election, we aggregate the weights to the cell-level. Each row now represents a unique joint distribution, with its appropriate weight. This reduces the size of the object to keep in memory by reducing the number of rows. Table 2 displays a subset of the completed frame. The frame is representative of 2011 demographics and 2014 voting behaviour. Given the fast-evolving demographics of India this is not optimal but reflects the most current data available.

## 4 Online convenience samples

The second critical component of our MrP election forecasting exercise implements a large sample survey of Indian voters. Most importantly, respondents are asked to indicate their vote preferences; we also include an extensive battery of covariate measures. This allows us to estimate the individual-level model that generates vote-choice predictions for the cells in our stratification frame. Typically, these are large N samples that approximate (often very roughly) national probability samples and hence provide some confidence that the predicted vote probabilities for the cells in the stratification frame are unbiased [1]. As we pointed out earlier, we will adopt an alternative sampling strategy for our training data. Our individual-level vote preference data will be generated from a decidedly non-representative convenience sample: not unlike the Xbox example referred to earlier, we will rely on post-stratification to deal with the non-representative sample characteristics. But unlike the Xbox example, we will rely on a comparatively smaller convenience sample—less than 7,000 vote intentions.

We demonstrate that a non-representative convenience sample of less than 7,000 vote intentions can generate reasonably precise and unbiased Indian vote predictions for our stratification cells. In India, like many other contexts, it is feasible to obtain, at a reasonable cost, large online convenience samples of respondents who can participate in a survey either on a computer, tablet or other personal device. These are not representative probability samples of the population which are virtually impossible in the India case but increasingly rare in most national contexts. These, quite unrepresentative, convenience samples, combined with a reasonable stratification frame, can produce accurate election forecasts.

We collect micro-level survey data from two different convenience samples of Indian voters. One of the convenience samples is an online subject pool recruited via Facebook

advertisements. We implement a quota sampling strategy via Facebook Ad Manager to approximate key demographic distributions. As a number of recent studies have pointed out, this can be both an efficient strategy for constructing non-probability samples of populations and one that results in reasonably accurate measures of public attitudes and behaviors [47, 48]. A second convenience sample consists of India-based workers from the Amazon Mechanical Turk platform. There is an extensive literature on sampling from Mechanical Turk, again suggesting that for many applications these convenience samples are fit for purpose [49, 50].

The survey questions reflect the model specification in Eq (1). Voter demographic profile questions are necessary for the cells of our stratification framed defined by the joint distribution of the values for *X*. In order to estimate vote turnout in 2019 (the *T* term in Eq (1)) we ask respondents to provide recalled 2014 vote turnout and their likelihood of 2019 turnout. Respondents are given the following question: *'This year, the General Election for the Lok Sabha is expected to be held sometime between April and May. Many people have told us they will not vote. How about you, do you think you will vote in the upcoming general election?'*. This question is clearly leading, and designed to curb significant over-reporting. The potential answers include 'Yes'; 'No'; 'Not sure at this point in time'; 'Not heard of upcoming election'. Missing values are treated as akin to 'Not sure at this point in time'. Our outcome variable in Eq (1) is 2019 vote choice ($\hat{v}$). We ask respondents to report their likely vote in 2019 and we also ask for their 2014 vote choice. Respondents are given the following question: *'If Lok Sabha elections were to be held tomorrow, which of the following parties would you vote for?'*. A list of options tailored to the state of residency is then provided, along with write-in options. Missing values are treated as akin to 'Not sure at this point in time'.

Online survey data collection began on February 20[th], roughly 13 weeks before the election date of the 19[th] of May; we concluded the surveys on April 10[th], the day before the beginning of the 6 weeks long election. S5 Fig in S1 Appendix shows the sample size per week to election, broken down by source and voting intention. Note that each of the major coalitions is composed of a myriad of parties; S1 Table in S1 Appendix describes allocations of parties to each coalition. The allegiances of a number of small parties changed over the course of the election resulting in expert disagreements on coalition allocations. Fortunately the alliance-changing parties tend to be small and ignorable at the national level, though they may well contribute to small discrepancies between this work and others. As of the 11[th] of April 2019, we had sampled 6, 785 voting intentions from 5, 807 users, where a unique user is identified using their IP address, and each IP address can respond to the surveys on multiple occasions; Mechanical Turks contributed 5, 152 responses from 4, 227 users; the Facebook-recruited subject pool contributed 1, 633 responses from 1, 580 users.

With the exception of caste, missing values on demographic covariates are rare—typically under 1% of the sample, with age-category being the worst offender at 1.5% missing. 10% of the respondents have missing caste, as a result of there being some surnames or caste subgroups (*Jati*) which we could not match to any of the 6 broader caste-categories used by the IHDS—the 6 caste categories use are: *Brahmin; Forward/General (except Brahmin); Other Backward Castes (OBC); Scheduled Castes (SC); Scheduled Tribes (ST); Others*. 12% of 2019 turnout responses and 10% of 2019 vote-choice responses are missing. Respectively, 32% and 37% of responses had missing values for 2014 turnout and voting behaviour. Missing values were an issue from February 20[th] to March 7[th], because we had not yet devised a method to post-stratify by past-vote, and hence had initially not asked past-behaviour questions.

We assume our NAs are missing at random, and use multiple imputation with chained equations, to again, as above, impute these via random-forest. Note we included an area-level predictor to make the imputations more robust; otherwise we use the same hyper-parameters as in the stratification-frame imputation model. Out of Bag (OOB) error is generated by

`missForest` and available in S3 Table in S1 Appendix. With respect to caste, 74% of OOB observations are correctly classified. 71% of 2014 turnout observations, and 83% of 2014 vote-choice are correctly classified; in contrast, 80% of 2019 turnout and 82% of 2019 vote-choice are correctly predicted. The chained random forests achieve reasonable classification-rates.

## 4.1 Turnout training data

The accurate measurement of turnout patterns at the deep-cell level is a challenge in traditional opinion polling under optimal conditions, and it becomes even more challenging in India, where traditional polling practices (i.e. simple random samples) are not viable. An accurate estimate of cell-level turnout weights is primarily of importance here because they translate into more robust estimates of national vote-shares that are conditional on turnout. Our ultimate forecasting goal is at the national level although deep-cell turnout estimates are an important contributor. We argue for two important features of these forecasts: a) the leveraging the INES turnout responses instead of those from our online pre-election survey; and b) we use the past-election turnout distribution to predict the present.

**4.1.1 Choosing a dataset.**   There is considerable evidence from pre-election polling studies that turnout over-reporting bias is significant. Analyzing survey data from the American National Election Study, Jackman and Sphan [51], estimate an upward bias in reported turnout of around 13 percentage points. They also document an additional source of turnout basis associated with non-response. In the UK, actual turnout is best approximated by only considering likely turnout by respondents who say they have a likelihood of turnout of 10, on a 10 point-scale (see https://ukpollingreport.co.uk/faq-turnout). Clearly, undecided voters who answer surveys are over-representing their chances of turnout. A prime reason pre-election surveys over-estimate turnout is social desirability bias [52]. Online pre-election surveys systematically under-sample non-voters; hence, turnout-bias cannot be mitigated by sampling-weight adjustments [53]. These factors have similar implications for our Indian Online convenience samples. As S5 Fig in S1 Appendix indicates, reported turnout in our online survey for 2019 is 96% which is highly inflated.

Researchers typically face imperfect options in selecting data than can inform estimates of actual turnout probabilities. To generate reasonable turnout measurement from pre-election surveys, the regression and post-stratification literature suggests using post-election face-to-face surveys from the most recent available general election to inform cell-level turnout weights [5]. Our less than perfect option to inform turnout probabilities for our Indian election forecast is the post-election 2014 INES turnout measure. As a rule pre-election forecasts should be conditioned on voter turnout probabilities. The precise value added of incorporating these turnout probabilities is of course an empirical question and very much depends on the quality of these estimated probabilities. Even in those cases, like our, where the turnout probability estimates are imprecisely estimated we demonstrate that there is predictive value added from including them in the modeling exercise. While the India NES also suffers from significant turnout over-reporting, it has a larger proportion of non-voters (9%) than is the case with the online surveys. Moreover, the INES is also a more representative survey than our convenience samples (see Table 1 in the next section).

A potential weakness in this approach, in addition to turnout over-reporting, is that 2014 turnout may not be representative of 2019 turnout patterns. And more generally, relying on past-election turnout, even if perfectly measured, may not accurately predict present turnout (it may work in our case but may not be generalizable to other elections). The data though suggest this approach is reasonable. First, empirically, this approach has been successful over multiple elections in other countries [5]. Second, with respect to India, we note the relatively strong

correlation (around 0.73—see S15 Fig in S1 Appendix) of turnout and lagged turnout, at the constituency level. This suggests that turnout behaviour of relatively small cells is consistent from election to election. Perhaps most importantly, past-election turnout is relatively unbiased (around 1% bias point over multiple elections) across constituencies, suggesting that as we aggregate cells to obtain national-level predictions, cell-level errors are likely to cancel-out.

In summary, the robustness of our national-level predictions very much depend on an informed cell-level turnout model. This requires selecting survey data on which to base this model. We opted for high-quality post-election turnout data from the 2014 INES over our more recent online convenience samples. The assumptions we make regarding the stability of turnout demographics over multiple elections seem plausible based on constituency-level analysis of historical turnout in India. We note here that an alternative approach to estimating turnout at the small-area level would involve relying on deep-level ecological inference enabled by the SHRUG dataset [54]. Relying on SHRUG data to predict turnout based on a historical high-resolution model holds the distinct advantage of generating predictions which do not necessarily suffer from upward-bias due to over-reporting, something that we have outlined as a significant limitation of models trained on survey data. Future work should attempt to develop methods to include ecological predictions derived from modeling SHRUG data in the turnout modeling strategy outlined in this paper; promising areas for exploration include weighted-averaging with our existing predictions as part of a deep model ensemble, or direct high-resolution multilevel modeling of turnout, including SHRUG covariates as part of an area-level predictor.

**4.1.2 Dealing with missing values.** Compared to our online pre-election polls, the INES has considerably fewer missing values. Demographic variables are mostly complete, with the exception of educational attainment (5% missing) and caste (9% missing); 2014 turnout has a mere 0.5% missing; and 12% of 2014 vote-choice is missing. Again we deploy `missRanger` to impute missing values and display the accuracy, as measured by OOB errors, in S3 Table in S1 Appendix. The imputation quality is acceptable: OOB education is correctly classified 71% of the time; Caste imputation is more noisy with a 57% correct classification rate; 2014 turnout is correctly classified 95% of the time; and 2014 vote-choice is correctly allocated 67% of the time. Considering the relatively small number of missing values, and the accuracy of the imputation algorithm with respect to turnout, we are confident in using the imputed dataset to train our turnout model.

**4.1.3 Accounting for sampling imbalance.** The resulting imputed turnout training data over-reports turnout. While this bias is the most severe, we also have evidence of other selection effects, as shown in the sample v. population plots available in the S1 Appendix. The low sampling variability of non-voters will significantly reduce statistical power and bias our cell-level estimates. We address selection effects by re-weighting observations. This is particularly important in our case because training a regularizing model, such as random forest, on an unbalanced sample of this sort can severely bias our conditional cell-level predictions due to shrinkage effects toward the global mean, This training imbalance is addressed by raking our turnout training dataset. We re-weight individual observations to ensure the sample is compatible with the marginal distributions for a number of target variables. In our case these include: 2014 turnout and vote choice, as well as demographic variables (gender and age), social variables (religion, caste, education and income) and geographic covariates (state).

## 4.2 Severity of non-representation

We get an indication of the extent to which the two convenience samples and the India NES, are not representative by comparing their distributions to those of the stratification frame. S6–

S14 Figs in S1 Appendix plot the distributions of a select number of variables in the stratification frame (our target population), against their convenience sample counter-parts. S6 Fig in S1 Appendix presents the population v. sample comparison for the distribution of 2014 voting behaviour; S7 and S8 present comparisons for the distributions of states and zones; S9 compares population and sample in their gender composition; S10 and S11 look at religion and caste respectively; S12 and S14 provide insights on the distributions of income and education, whilst S13 presents the comparison between population and sample in the distribution of age categories. Note that we employed sampling constraints based on zone, age and gender in the later stages of sampling for the Facebook pool which resulted in a more representative pool compared to the Mechanical Turks, though these constraints seem to have been more effective on geography than gender or age. Fig 1 showcases the degree of 'representation' for the two convenience samples and the INES across a number of marginal distributions (these are disaggregated in the S1 Appendix).

With respect to geographic distribution across the samples (S7 and S8 Figs in S1 Appendix), our work confirms previous findings [55, 56] regarding the extreme over-representation of Tamil Nadu and Kerala; together they make up only 9.8% of the voting age population in our stratification frame and yet represent over 80% of Mechanical Turks. Geographic over-representation in the Facebook sample is not severe, partly as it is controlled by sampling quotas, and partly because the geographical distribution of Facebook users is more representative of the population [55]. The distribution of states in the 2014 INES mirrors that of the population.

Gender (S9 Fig in S1 Appendix) is properly balanced in the INES. Both convenience samples over-represent males—with Mechanical Turks being further skewed toward males than Facebook users. Religious affiliation in our convenience samples is similar to that in the stratification frame, though unsurprisingly due to the geographical skew in favour of Kerala and Tamil-Nadu, Christians are over-represented amongst Mechanical Turks (S10 Fig in S1 Appendix). With respect to age, the INES is representative of the population at large while the convenience samples are biased: Mechanical Turks are overwhelmingly between 25 and 35 years old, while Facebook users are even younger, exhibiting over-representation of all under 35s (S13 Fig in S1 Appendix).

Perhaps unsurprisingly, Brahmins make up a plurality in both our convenience samples (S11 Fig in S1 Appendix). We heavily over-represent individuals in the 'Other' caste category which includes people who do not subscribe to caste system, refused to be labeled or explicitly selected an 'Other' option (a response significantly more common amongst Facebook users). Individuals from 'Scheduled Tribes' are under-represented. The distribution of caste in the INES, on the other hand, seems to over-represent Backwards Castes.

The INES best approximates the population in terms of yearly household income (S12 Fig in S1 Appendix) while the Mechanical Turk and, in particular, the Facebook samples are less representative on income. Mechanical Turks are richer than the voting-age population: 16% of the sample is in the top two income brackets. Facebook users are overwhelmingly high-income, with 59% of the sample clustered in the top-two income categories. In the reference population 1.6% are concentrated in these top-two categories. Education levels exhibit the most severe bias (S14 Fig in S1 Appendix). The INES over-samples individuals with no formal education, but otherwise displays a reasonable distribution of education levels. The two convenience samples are both made up largely of graduate-educated individuals: 92% of our Mechanical Turks have some graduate education, in contrast with a striking 95% of our Facebook sample and a mere 8.9% of the population at large.

Regarding voting behaviour in the last election, the 2014 INES best approximates the vote, though again significantly over-reporting turnout levels (S6 Fig in S1 Appendix). Individuals from the Facebook group are significantly more likely to have voted for the NDA in 2014, and

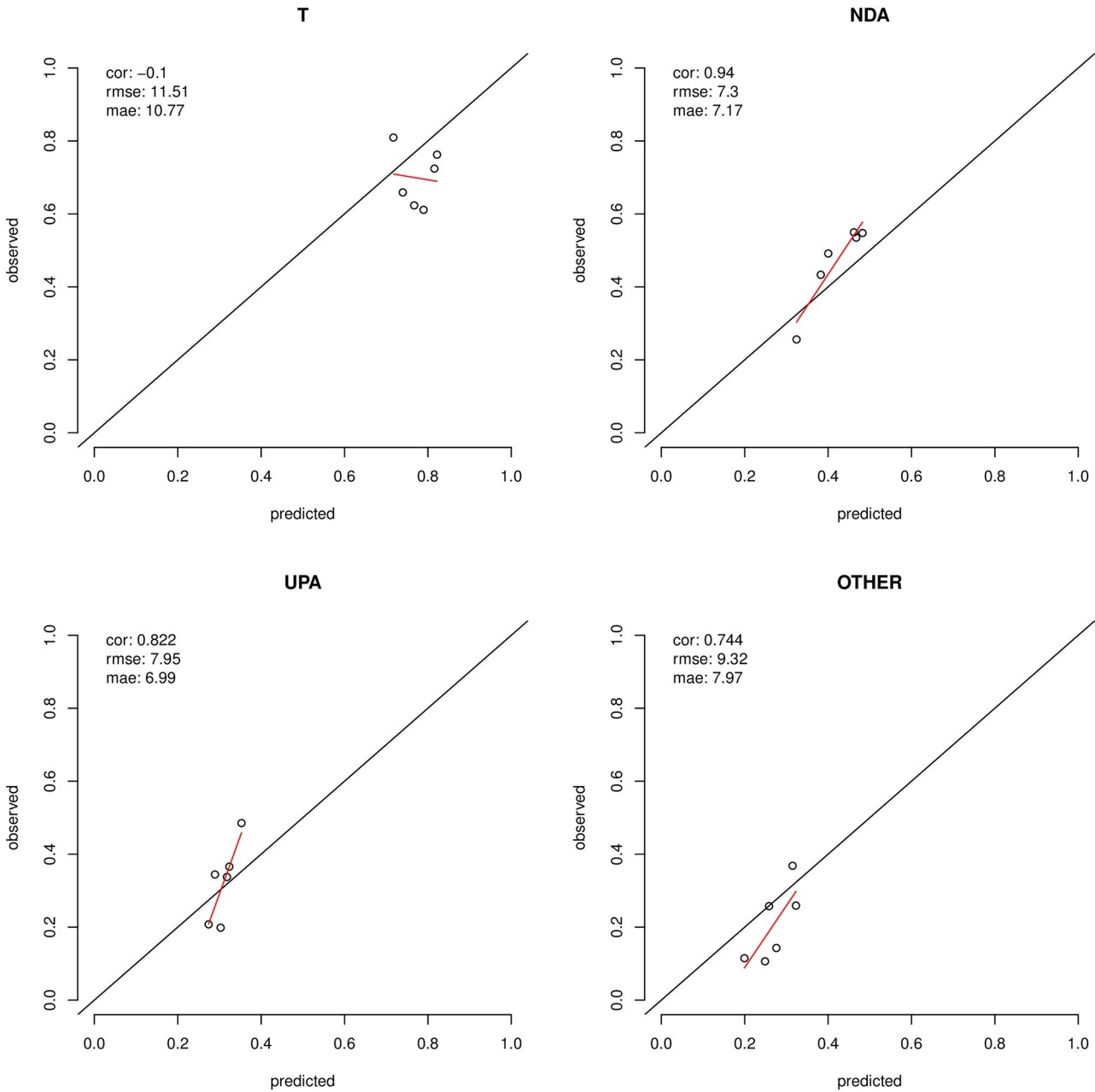

**Fig 1. Summary plot assessing the degree of bias, when compared to the estimated stratification frame, across a number of marginal distributions, in each of the three samples used in the analysis.** The most severe discrepancies are highlighted. Each dot represents the % of an attribute, such as education level, caste or income, in the population as a whole and in the sample at hand. If the sample at hand is perfectly representative of the population of interest, the dots should lie on the $y = x$ line.

significantly less likely to have voted for the UPA. Mechanical Turks are somewhat more likely to have voted NDA in the last election, though they are also more likely to have voted for the UPA—and significantly less likely to have voter for a third alliance. Turnout behaviour is exceedingly unrepresentative for both convenience samples, with 97% of all respondents saying they turned out to vote, compared to 66% of eligible voters in 2014, reflecting both a higher

degree of political engagement due to their demographics, but almost certainly a high degree of over-reporting.

In summary, our two, non-representative, convenience samples exhibit similar biases: these samples are more male, younger, higher-status, highly educated, and highly politically engaged. This is consistent with most online convenience samples of this nature that require online literacy and English language proficiency. There are mode-specific selection effects. The work-environment provided by the Mechanical Turk platform results in participants who are more diverse in their income; less diverse in their age-cohort; less diverse geographically. This reflects to some extent an uneven penetration of desktop internet access. The leisure and connectivity provided by Facebook results in more geographically diverse participants from multiple age-cohorts and more balanced gender representation. These Indian Facebook sample characteristics are similar to those documented in other Facebook recruited samples from developing countries [47]. And overall, our findings are consistent with previous Indian studies employing online sampling [55–58].

## 5 Modeling behaviour

Having built a reliable post-stratification frame and having collected relevant training data for turnout and vote-choice, we now model behaviour at the cell-level. Our goal is to aggregate post-stratified cell-level behaviour. Our national point estimates for turnout are calculated from Eq (2) and our national estimate of vote-choice conditional on turnout from Eq (3):

$$\hat{\tau} = \frac{\sum_i \hat{\Pr}_i(T \mid X) \times \hat{N}_i(X)}{\sum_i \hat{N}_i(X)};$$ (2)

$$\hat{v} = \frac{\sum_i \hat{\Pr}_i(V \mid T, X) \times \hat{\Pr}_i(T \mid X) \times \hat{N}_i(X)}{\sum_i \hat{\Pr}_i(T \mid X) \times \hat{N}_i(X)}$$ (3)

In Eqs (2) and (3), $i \in 1, \ldots, N$ is an index of cells, where each cell represents a row in our stratification frame; $X$ denotes the covariate characteristics that uniquely identify a cell; $N_i(X)$ is the number of individuals belonging to cell $i$ in the population; $Pr_i(T \mid X)$ is the turnout probability of each cell, conditional on the cells' characteristics; and $Pr_i(V \mid T, X)$ is the vote-choice density of each cell, conditional on turnout and the cells' characteristics.

While $\hat{\Pr}_i(T|X)$ is estimated using exclusively the 2014 NES data, $\hat{\Pr}_i(V|T, X)$ is actually taken to be the simple average over our convenience samples $s \in \{MechTurk, fb\}$:

$$\hat{\Pr}_i(V \mid T, X) = \frac{\sum_s \hat{\Pr}_i(V \mid T, X, S = s)}{\sum_s 1\{S = s\}}$$ (4)

There are two relevant types of uncertainty around these point estimates: First there is parametric uncertainty related to the point-estimates of conditional turnout and vote choice distributions. A second is sampling uncertainty, that, conditional on a given estimate of the conditional densities, accounts for variance across random draws of the turnout and vote-choices for each cell. Eqs (7)–(9) propose a strategy for characterizing the uncertainty in our model by simulating the election, $D$ times, where separate random draws are indexed by $d = 1$,

..., $D$, according to a probabilistic model:

$$T_{d,i} \sim \text{Binomial}(\text{Pr}_{d,i}(T \mid X), \hat{\text{N}}_i(X)); \tag{5}$$

$$V_{d,i} \sim \text{Multinomial}(\text{Pr}^{\star}_{d,i}(V = j \mid T, X, S), T_{d,i}); \tag{6}$$

$$\text{Pr}_{d,i}(T \mid X) \sim \text{Beta}(\mu_i^T, \sigma_i^T); \tag{7}$$

$$\text{Pr}_{d,i}(V = j \mid T, X, S) \sim \text{Beta}(\mu_i^{V=j}, \sigma_i^{V=j}); \tag{8}$$

$$\text{Pr}^{\star}_{d,i}(V = j \mid T, X, S = s) = \frac{\text{Pr}_{d,i}(V = j \mid T, X, S)}{\sum_j \text{Pr}_{d,i}(V = j \mid T, X, S)} \tag{9}$$

Relevant probability densities are assumed to behave according to a series of independent Beta distributions, each fully specified by a mean and a variance parameter. Vote choice quantities are also assumed to be independent, and normalized a-posteriori. The objective of our estimation strategy is to obtain estimates for $\mu$s and $\sigma$s. By aggregating each set of simulations $d$ as indicated in Eq (9), we recover an empirical density for aggregated vote-choice and turnout. We can then make probabilistic statements about the election results.

## 5.1 Multivariate outcomes

The parametric density $Pr_i(V|T, X)$ is multivariate in nature—that is, we are interested in predicting the vote-choice probabilities for individual $i$ with respect to each option $j$ = {*NDA*, *UPA*, *Other*}. To estimate these quantities we rely on the *one v. all* [59, 60] strategy for predicting choice-probabilities for each category. This involves normalization of the resulting scores into probabilities via either standard normalization, or a more complex multivariate link function such as *softmax* [61]. In our Lok Sabha study, given we are dealing with a single layer of predictions for three alliances with similar levels of popularity, we have no reason to think employing softmax normalization will provide any advantage over standard normalization. Softmax is especially relevant if we encounter corner-probabilities, as it will tend to add weight to probabilities close to 1, and down-weight probabilities close to 0, relative to the other components one is normalizing over. The context where it is most useful is in multi-layer neural networks for classification, where it works by increasing the probability of extracting strong and definitive signals over noise. Hence we resolve to normalize the predicted mean of the vote-choice densities as shown in the Eq (9).

## 5.2 Choosing a learner

We employ Random Forests [8, 62] to estimate $Pr_i(V \mid T, X)$ and $Pr_i(T \mid X)$. RFs are a *bagging* algorithm (also known as *bootstrap aggregation*) combined with an *ensemble* method [12]. The bagging algorithm iteratively takes a large number of bootstrap samples of the training records and features in order to fit, ideally, uncorrelated CARTs. The ensemble feature refers to its predictions being a simple average of the bootstrapped CARTs. RFs are optimized for speed, especially in the `ranger` implementation [9], as they can rely on parallel processing and other computational features to generate trees of optimal depth. RFs are amongst the most consistent performers in terms of their ability to produce high-quality predictions in various competitions. Some researchers claim *'the random forest is clearly the best family of classifier'* [63], although this is contested [64].

Eq (10) generates our RF predictions. The RF predictor is the mean behavior of a given cell $i$; $\mu_i$, is the the average of $B$ regression trees, each fit to a random sample of training data sets $T$ $(y_o, x_o)$, $\forall o = 1, \ldots, O$, where $o$ indexes the respective training observations, according to a set of random parameters $\Theta = \{\theta_1, \ldots, \theta_B\}$:

$$\mu_i = \frac{1}{B} \sum_b \phi_{i,b}(x, \theta) \qquad (10)$$

where $\phi_{i,b}$ stands for the prediction of component tree $b$ for cell $i$.

As we pointed out earlier, we can also estimate $Pr_i(V \mid T, X)$ and $Pr_i(T \mid X)$ using multilevel regression [65]. Conventional MrP has performed well for many election forecasts. In particular its strengths can be condensed to the following: its ability to *regularize* predictions ("borrow strength" from aggregate-level data) [66] and to incorporate level-specific heterogeneity in the modeling; its parametric nature, which makes multilevel regression highly interpretable, when compared to non-parametric black-box models; its ability—especially if estimated under a Bayesian framework—to incorporate uncertainty in the predicted quantities and estimated parameters. As with any estimation strategy, it has its limitations: MrP predictions can be hugely sensitive to model specification, potentially hindering its performance in cases where not much is known about an underlying functional form; attenuation bias [45] due to over-regularization can reduce the accuracy of small-area estimates by shrinking these toward the global mean; multilevel regression models do not scale—under current computational constraints—beyond moderate sample sizes. Monte-Carlo estimation of Bayesian Multinomial models *does not support large N*—current state-of-the-art applications for the UK take 12 hours, for around 100, 000 responses. The MrP literature has began to address these challenges by moving towards more semi-parametric modeling: Sparse regression [67] is a proposed solution that enforces automated variable-selection. Variational inference [68, 69] improves scalability but at the cost of typically biased uncertainty estimates. Regression and post-stratification practitioners have also turned to machine learning: Bayesian Additive Regression Trees (BART) [70] have been shown to out-perform regular MrP in some applications. Random Forests [7] have been used to deal with high-dimensional settings. Stacking [71] of several non-linear learners shows promise, though it too suffers from computational speed issues.

We also recognize that the advantages of the RF estimation we employ here are sensitive to the data and modeling goals. Our Random Forest estimator deploys automatic variable selection, adaptive shrinkage and non-linear modeling. These can enhance modeling performance when sample sizes are relatively large, which is the case here since we have approximately 7, 000 observations in our vote-choice model, and over 14, 000 in our turnout model. With smaller public opinion samples, *non-linear effects are limited* and additive functional forms are well-understood, limiting any gains from machine learning. The principal objective of this modeling exercise is correctly predicting national election outcomes. Calibrating the effect size of covariates is not a major concern. Machine learning methods are not ideal for modeling exercises in which the interpretation of effect sizes is paramount. Some learners—such as Classification and Regression Trees (CART) [72]—retain intuitive interpretation. The literature has developed measures of importance and significance of the input features [73, 74] for more complex algorithms but they typically are not intuitive.

The *prediction uncertainty* associated with RF modeling has only recently received serious attention. Cross-validation [75] is the industry-standard for obtaining uncertainty estimates around machine-learning predictions. This assumes that the training sample provides a deep and detailed picture of the population of interest. This might be a plausible assumption under big-data conditions but it quickly breaks down for relatively small convenience samples.

Recent advances in the statistical understanding of bagging estimators are a promising direction for improving assessments of prediction uncertainty [76–78].

## 5.3 Hyper-parameter tuning

RFs are ensembles of CARTs, and as such are governed by numerous hyper-parameters which describe the tree-growing mechanism. Unless otherwise specified, we use the same set of hyper-parameters for both our turnout and vote-choice predictions. We set `num. trees = 1500`, hence growing a large-enough number of trees to ensure no Monte-Carlo effects are present in our estimates. We set the number of candidate features to split, `mtry = p`, where $p$ is the total number of available features, hence requiring that all features be considered for splits in each tree. This effectively transforms our RF to an ensemble of bagged CARTs. We do this in order minimize the shrinkage effect of the forest. Building on [11] we prioritize a model that allows for cross-level interactions to emerge in each tree. If these interactions explain residual variance and if trees are precluded from considering these interactions because a subset of the relevant variables is randomly out-of-bag, each tree will be a weaker predictor, and the ensemble will be weak as a result. Furthermore, the recent literature on growing probability-forests [79] explicitly suggests a large `mtry`—and in particular mtry = p outperforms other tuning choices in some key UCI machine-learning repository datasets [80]. In principle, we could have set `mtry` to any arbitrarily large number; however we found that bootstrapped trees, which are the conceptual equivalent, to be more attractive, as they represent a distinct, well-defined modeling strategy in the literature [12, 62]. On the other hand, having each tree consider the same covariate space may increase the correlation across trees, and decrease the ensemble predictive gains [12]; as a result, we reduce the size of each bootstrapped sample, `sample.fraction` $= \frac{1}{3}$ and impose sampling without replacement, `replace = FALSE`, such that each record in each bootstrap sample is unique, and the samples are small-enough that they are extremely likely to be highly heterogeneous amongst themselves. Note that this will also help reduce fitting time. No limit is explicitly imposed on the depth of each tree, `max.depth = NULL`. The tuning of `min.node.size` is dependent on the complexity of the dataset at hand: for a relatively well balanced dataset, the default value for a probability machine is assumed to be `min.node.size` = 0.1×$N$, where $N$ is the size of the training data [81]. However, our own experimentation revealed this value is inadequate for imbalanced datasets, as predicted values are excessively regularized by the dominance of the most likely outcome in the terminal nodes. Hence, to obtain meaningful differences in cell-level predictions, we set `min.node.size = 30` for the turnout model, and respect Malley et al.'s advice for the vote-choice model. The choice of the value 30 is arbitrary, but driven by the need to have large-enough terminal nodes such that they don't collapse entirely onto 1 or 0, but rather provide meaningful probability values, as well as the need to relax node-size-driven regularization to increase heterogeneity in predictions. S20 Fig in S1 Appendix shows a reasonable degree of heterogeneity for mean predicted turnout across cells in the stratification frame, though a clear upward bias persists.

## 5.4 Prediction error

Wager and Athey [76, 77] have shown that Random Forest predictions are asymptotically Gaussian in typical regression settings, and provide a consistent estimator of the asymptotic variance via the infinitesimal jackknife procedure [82]. Lu and Hardin [78] have instead developed a consistent estimator of the conditional prediction error for RFs, leveraging the OOB observations in each tree. The simple version of their procedure is highly intuitive: for every cell in the test-set, they go through each tree in the forest and record its out-of-bag neighbours

—observations which were randomly not chosen to train the given tree, but were predicted to be in the same terminal-node as the cell-observation. They then calculate the error for each of these neighbours; since their true observed outcome is available it can be matched against the prediction for each tree. They then take a weighted average of this error based on the number of times each of these observations is an out-of-bag co-habitant of the test-cell. The resulting score is the conditional MSPE.

We should note here that, to our knowledge, no estimator for the prediction error of probability machines—i.e. models which output predictions in the range [0, 1]—has been thus far presented in the literature. The Lu & Hardin estimator does not impose a functional form over the distribution of the error term, hence it can be directly employed to parametrize non-gaussian distributions with [0, 1] support, which are also indexed by some measure of MSPE. The Beta distribution is such a density, as it can be fully parametrized by its mean and variance, as suggested in Eqs 7 and 8 earlier in this section.

It follows from the definition of the Beta distribution that: i) $\mu_i^T \in (0, 1)$ and $\mu_i^{V=j} \in (0, 1)$; ii) $(\sigma_i^T)^2 < \mu_i^T(1 - \mu_i^T)$ and $(\sigma_i^{V=j})^2 < \mu_i^{V=j}(1 - \mu_i^{V=j})$. In our application, the first set of conditions regarding the mean holds for the entirety of our RF predictions; the second set—the conditions relating to the variance parameter—are more problematic. S16 Fig in S1 Appendix illustrates the effect the truncation has on the distribution of mean turnout and mean vote-choice across cells. The red dots in the plot indicate predictions whose MSPE exceed the beta-imposed limit. We see that these out of limit predictions are a small proportions of the total cells, for each predicted quantity. The limit imposed by the distribution is typically close to the estimated MSPE and is only really a problem for means closer to the corners of the distribution; hence we do not expect the correction to the limit to have significant consequences on our estimates. Having assumed this functional form, we can simulate $D = 500$ draws from our probabilistic model and obtain credible uncertainty estimates around our cell-level behavioural predictions. These can be aggregated by each simulation draw to preserve uncertainty at the national-level.

The MSPE estimator is implemented with the `forestError` package for R which can be computationally expensive. To limit computational costs, we randomly sample 1% of the cells in our stratification frame—an arbitrary amount large-enough to contain a reasonable amount of heterogeneity, but small enough to limit the computational burden. We then run the `forestError` algorithm on this smaller data set in order to derive their conditional MSPE. These predicted errors are used as training data to estimate the error of the remaining cells via RF regression implemented by `ranger` using the default hyper-parameters. This provides an approximate measure of uncertainty for each cell in the stratification frame. Note that our practical problem is to obtain reasonable confidence bounds around national-level estimates of vote-share, and as a result we can afford to be imprecise at the cell-level, as we expect errors to cancel-out by aggregating simulations at the national level.

## 6 Results

The empirical test for our RF Post-stratification estimation approach is a forecast of the 2019 Lok Sabha election (that we published before the close of balloting). We assembled data that reflected the realities of developing countries like India: a stratification frame augmented by multiple data sources and an unrepresentative online convenience sample. Our conjecture is that these imperfect data combined with RF post-stratification estimation can generate accurate turnout and vote choice predictions. We generate these at the national level and also at two sub-national levels, the zone and the state. Our results suggest that reasonable levels of prediction accuracy can be obtainable from unrepresentative online convenience surveys. We

then apply a uniform swing model [83] to obtain predicted seat-counts and show that our predictions outperform other available polling. Note that the election unfolds over 5 weeks, and we stop monitoring right before the beginning of voting, hence our point estimates refer to our predictions roughly 5 weeks out from the final election day.

## 6.1 National behaviour

Fig 2 shows the proportion of individuals who would turn out to vote over the course of the voting period. These predictions are based on the NES 2014 training data, after re-weighting to control for observable selection, and assuming stable turnout dynamics over demographics. Our point estimate is 78.9% against an observed turnout of 67.5%—a difference of 11.4%. This is in line with upward turnout bias observed in other countries in high-quality surveys [51]. The three standard deviation prediction uncertainty over the mean is just over ±1, which is rather narrow.

Fig 3 shows the national vote share predictions over the course of the campaign. Once voting began, we stopped collecting survey data and our predictions are simply projected forward according to a random walk. As the five-week voting period began, our predicted point estimate for the NDA is 41.8%, against an observed value of 45.0%—contributing to a prediction

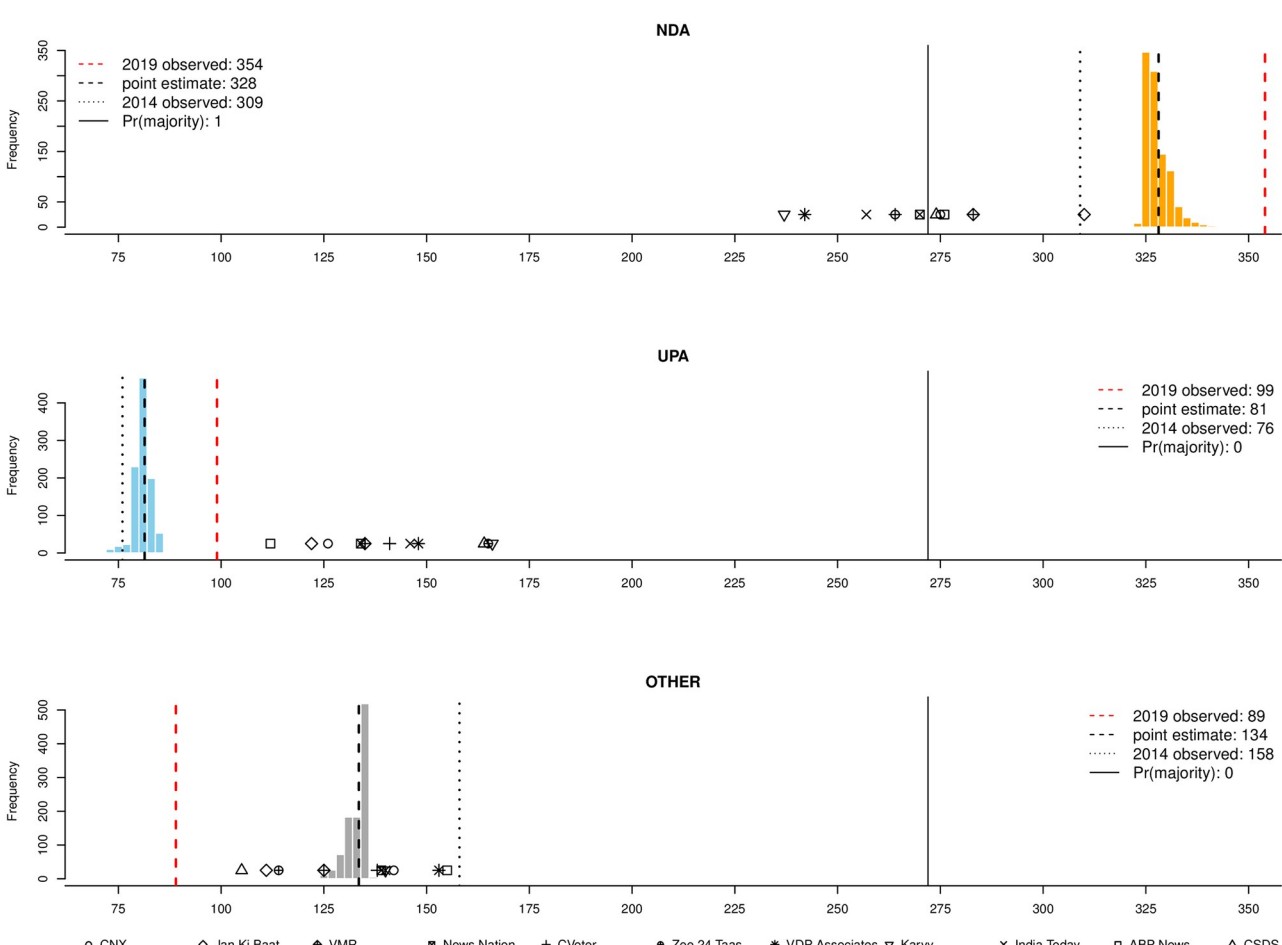

**Fig 2. Predicted distribution of turnout at the national level.**

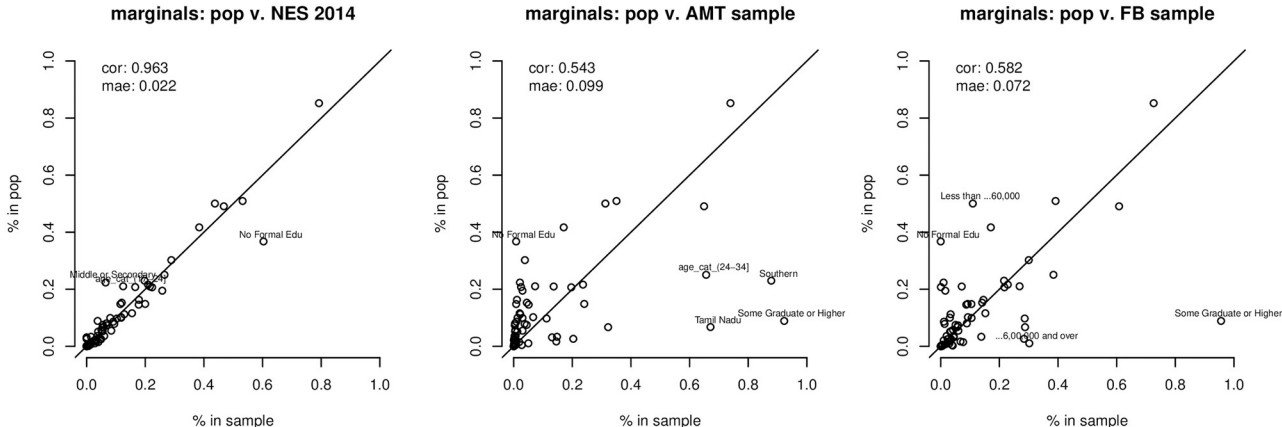

**Fig 3. National vote share for the three major alliances over the course of the campaign; monitoring stops before the beginning of voting, and vote-share after this point.**

error of 3.2% points. The point estimate prediction for the UPA is just under 30.8% against an observed vote share of just under 31.3%, contributing to a prediction error of around 0.5%. Finally, the remaining error on the 'Other' alliances is just under a 3.7% over-estimation. On average, we have a prediction error of 2.5%, which is well within the range of standard polling error using traditional polling methods [84]. With respect to uncertainty, again we see a three standard deviation interval of a very narrow ±1. Projecting our estimates to the last day of the election, according to a random walk, the 3 s.d. prediction intervals for NDA, UPA, and 'Other' alliances expand respectively to ±5.1, ±5.3 and ±7.4 points. The uncertainty over the results on the last day of the election, obtained by projecting the last estimate of vote-choice forward according to a random walk, ends up being quite large, which reflects the fact that voting behavior over the course of the monitored weeks was rather variable from week to week.

It is important to understand whether the positive performance outlined above is attributable to the quality of the online pre-election polling, or whether it is the result of our modeling strategy. S4 Table in S1 Appendix presents the national-level predictions at 6-weeks to the final election day, derived from the un-modeled, raw online sample. It is clear that modeling adds value: the national NDA vote share predicted from the raw sample amounts to 50.8% (error = −5.8%), a more severe bias, though in the opposite direction, compared to the modeled prediction; the UPA raw vote share was rather reasonable, at 32.4% (error = −1.1%), though the 'Other' alliances are severely under-valued, at 16.8%(error = + 6.8) points. This analysis shows that modeling and post-stratification is worth around 2 percentage points in error reduction per alliance on average. The value of modeling will become even clearer in the seats predictions phase, as the large over-estimation of the raw NDA share would lead to a collapse of the UPA according to a uniform swing prediction.

It is further worth highlighting the role that turnout modeling plays in error reduction— put plainly, to what extent is our prediction more accurate as a result of modeling conditional on turnout, and to what extent is the turnout modeling architecture justified? These questions are answered in S21 Fig in S1 Appendix, where the national projections from a model which completely ignores turnout are presented. A model which ignores turnout weights and relies only on vote-choice of online respondents to monitor public opinion would have an average error per alliance of 3%, about 0.5% higher per alliance on average. The turnout modeling seems to affect the alliance levels of support, and works specifically by reducing the support for

'Other' alliances, and increasing the support for the UPA; not much of a change can be reported relative to the NDA.

Another element of our modeling effort which is worth highlighting is the ability to capture real-time shifts in public support due to salient campaign events. We focus on the spike and subsequent stabilization of support in favour of the NDA from Fig 3, between the beginning of measurement and 9 weeks to the final day of the election. This spike corresponds to a military skirmish that occurred in February and March between India and Pakistan, which saw the NDA and its leader Narendra Modi styling themselves as 'Watchmen', highlighting the NDA's focus on national security [85]. The ramifications of the border skirmishes with Pakistan on Indian public opinion are well captured in our public-opinion estimates, and amounted to a pro NDA shift worth just under 2.3% at the height of the tensions. Repeated low-cost surveying of online convenience samples, and subsequent modeling adjustments, enable the estimation of such real-time preference-shifts.

## 6.2 Feature importance

There are hypothesis tests that assess the contribution that different independent variables make to RF predictions [73]. We implement a permutation *variable importance metric* (VIM) [86, 87] that indicates whether our priors (such as the correlation between age and turnout) are being learned and leveraged in some way by the algorithm. We also use VIM to rank variables according to their predictive contribution.

The VIM score is calculated by measuring the change in average OOB predictive accuracy when each feature is rendered more stochastic via random permutation. Larger changes in predictive accuracy provide an indication of the relative importance of a predictor variable. Non-parametric hypothesis testing can then be performed by first simulating a distribution of the predictors' VIM under the null hypothesis that the predictor is 'unimportant'. The null distribution is obtained by randomly permuting the response variable $P$ times, and recording the associated 'null importance' of the features for each permutation. We can then simply calculate the percentage of null-scores that are greater than the observed VIM. When strong predictors are correlated, such that there can be clusters of highly predictive correlated features, the Altmann permutation VIM [86] distributes importance across correlated features, and assigns significant p-values to all of the predictive features in the group. We should not interpret the effects independently but rather report the total effect with their group-cohabitants. S17 Fig in S1 Appendix presents the most significant feature importance scores for each of our behavioural models.

Beginning with our turnout model, we can see the random forest picks up a large number of significant important variables—though a large number of them have seemingly little to no impact on the prediction if taken independently. This is in part a direct result of the low `min.node.size` we specified, which encouraged the forest to explore a large number of statistically meaningful—but small in magnitude—deep interactions. A few variables stand-out as being responsible for a meaningful magnitude of error reduction: individual-level alliance of choice is the most impactful variable for making our turnout predictions, followed by religious affiliation, area-level age distribution, caste, income and education level. The model therefore points to three broad classes of determinants of turnout: political preferences; age distribution and class.

With respect to 2019 vote choice, 2014 vote is the most predictive variable across all alliances. Caste is also a significant determinant of vote-choice. Religious affiliation plays a key role in the UPA vote, perhaps indicating the relatively strong Muslim affiliation with the alliance. State-level caste-composition and age profiles are also significant. Many distance

variables, that measure geographic proximity, are significant across alliances. This indicates there are cross-state correlations and geographical patterns in the vote resulting in a degree of local smoothing. Weeks to the end of the election play a significant role for all alliances which suggests significant campaign effects. There are more significant predictors of the vote for the NDA than for other alliances. The 'Other' group is the least predictable (low VIM magnitude per variable and little significance across these), perhaps a sign of the heterogeneity of the 'Other' category which contains many diverse regional and national parties.

## 6.3 Sub-national behaviour

Typical MrP applications have as primary objective the estimation of behavioural outcomes at the sub-national level [65]. Our application is different, in that we use regression and post-stratification simply as a way to produce more robust national-level forecasts from non-representative surveys. Nevertheless, we explore the degree to which relatively small and unrepresentative surveys can be smoothed to predict sub-national behaviour in this highly heterogeneous context. We include two geographical levels of interest: zones and states; there are 6 zones and 35 states in our application. Though Telangana formally seceded from Andhra Pradesh in 2014, we ignore this split and still consider these two states as one. This allows us to avoid having to adjust census and IHDS population estimates for a state which at the time was not recorded. Fig 4 and S18 Fig in S1 Appendix compare our predictions at the sub-national level with the observed election results.

Overall, sub-national accuracy is poor. We expected this to be the case, especially given the highly skewed and unrepresentative geographic distributions of our survey data. Zone-level predictions outperform state-level ones significantly, showing reasonably high cross-zone correlations for the two major alliances. In particular, state-level forecasts for the NDA vote are correlated with the observed outcomes at 0.94, suggesting a reasonably strong sub-national signal was present in the pro-NDA survey response. The poor performance of our predictions at the sub-national levels suggests a lack of area-level statistical power. In circumstances where there is high cross-area correlations, as is often the case in US or UK, sub-national estimations, area-specific statistical power is less of an issue [1] than it is in India, where cross-state heterogeneity in political behaviour can be dramatic.

## 6.4 Seat projections

Our forecasting method allows us to 'call' the election in terms of seats. We assign a predicted number of seats to each alliance and measure the probability of obtaining a majority in the Lok-Sabha. Our national-level vote-share estimates are converted into seats by applying a "uniform swing" [83, 88]. In this model every constituency 'swings' with the national vote, such that a national change from the last election in the vote-share of each party will be reflected in an identical change at the constituency level. More formally, we can estimate the 2019 constituency-level vote share of a given party as follows:

$$\hat{\delta}_{j,2019} = \hat{v}_{j,2019} - v_{j,2014}; \tag{11}$$

$$\hat{\lambda}_{j,2019,c} = \lambda_{j,2014,c} + \hat{\delta}_{j,2019}; \tag{12}$$

where $\delta_{j,t}$ indicates the change in vote share for a given party, for an election held at time $t$; $\lambda_{j,t,c}$ is the vote share for party $j$ at time $t$ in constituency $c$, for $c = 1, \ldots, C$.

Under this model, a specific constituency may be poorly predicted, but the headline share of seats tends to be accurate. This results because the true constituency swings tend to be

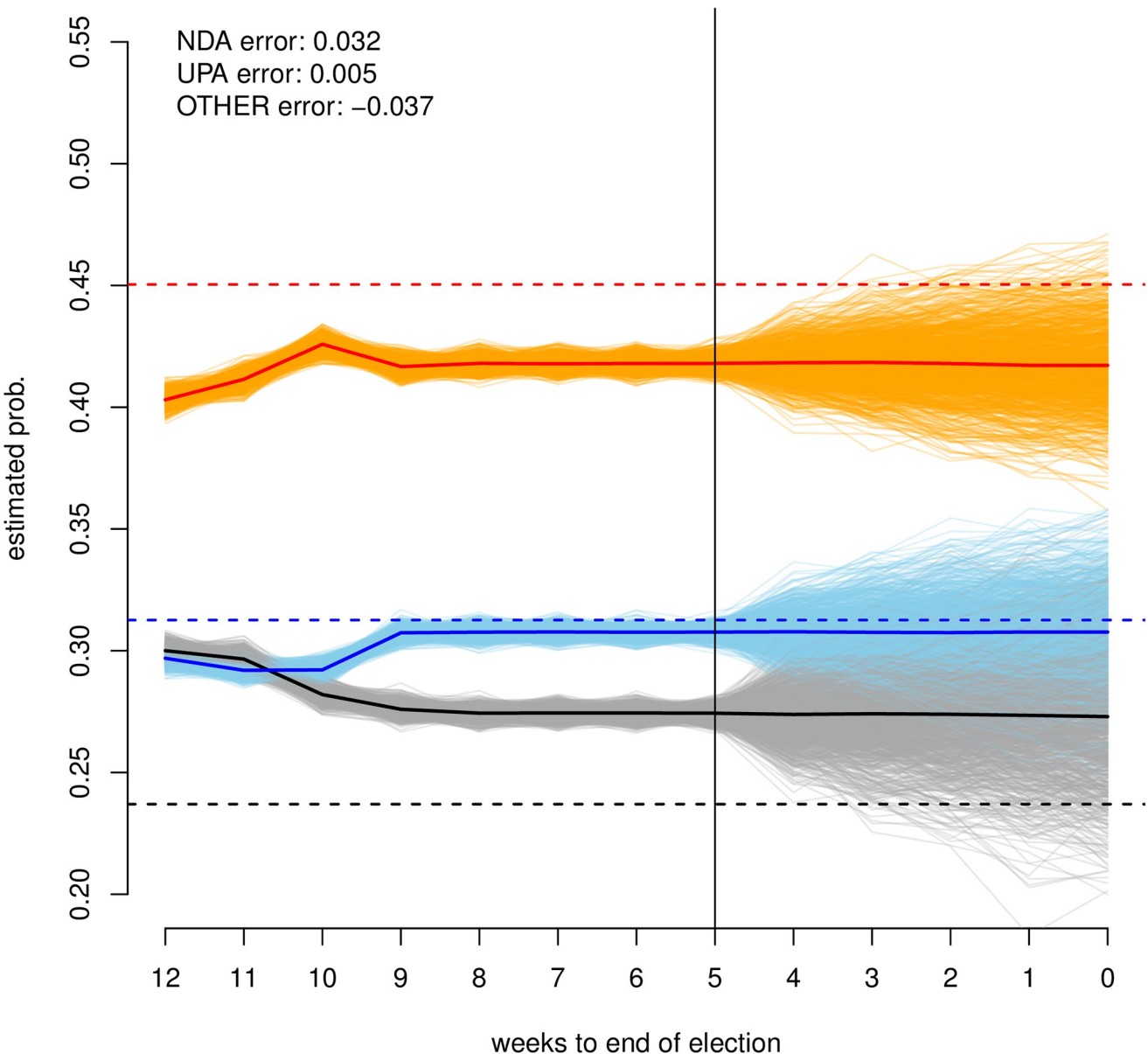

**Fig 4. Zone-level predictions v. observed zone-level 2019 behaviour for Turnout share (top-left), and Alliance vote-share (NDA top-right, UPA bottom-left, Other bottom-right).**

realizations of a Normal distribution with the national-level swing as its mean. At the national level, the uniform swing model tends to work quite well in the US and the UK due to the relatively uniformity of the political climate (meaning political discourse is common across constituencies); the relative homogeneity of constituencies in terms of important socio-economic variables such as class and income; and the relative stability of the voting choices. While India is a far more heterogeneous country, there is evidence that a uniform swing may apply [83,

**Table 3. Seat projections and errors across pollsters.** Performance of two baseline predictions (the past-election results and the uniform-swing projection from the raw, non-modeled online data). Underlined are the best performing pollsters for each column. In some case we present two bold quantities per column because the best performing was so outdated—we highlight a second-best performing poll that was more recent and comparable.

| House | Days Left | NDA Pred. | Error | UPA Pred. | Error | Other Pred. | Error | Abs. Error | NDA—UPA Pred. | Error |
|---|---|---|---|---|---|---|---|---|---|---|
| 2019 Result | | 354 | | 99 | | 89 | | | 255 | |
| Cerina & Duch | 39 | 328 | 26 | 81 | 18 | 134 | −45 | 89 | 247 | 8 |
| Cerina & Duch (raw) | 39 | 388 | −34 | 26 | 73 | 129 | −40 | 147 | 362 | −107 |
| 2014 Result | | 309 | 45 | 76 | 23 | 158 | −69 | 226 | 233 | 22 |
| CNX | 43 | 275 | 79 | 126 | −27 | 142 | −53 | 159 | 149 | 106 |
| Jan Ki Baat | 45 | 310 | 44 | 122 | −23 | 111 | −22 | 89 | 188 | 67 |
| VMR | 49 | 283 | 71 | 135 | −36 | 125 | −36 | 143 | 148 | 107 |
| News Nation | 49 | 270 | 84 | 134 | −35 | 139 | −50 | 169 | 136 | 119 |
| CVoter | 49 | 264 | 90 | 141 | −42 | 138 | −49 | 181 | 123 | 132 |
| Zee 24 Taas | 49 | 264 | 90 | 165 | −66 | 114 | −25 | 181 | 99 | 156 |
| VDP Associates | 80 | 242 | 112 | 148 | −49 | 153 | −64 | 225 | 94 | 161 |
| Karvy | 109 | 237 | 117 | 166 | −67 | 140 | −51 | 235 | 71 | 184 |
| India Today | 139 | 257 | 97 | 146 | −47 | 140 | −51 | 195 | 111 | 144 |
| ABP News | 200 | 276 | 78 | 112 | −13 | 155 | −66 | 157 | 164 | 91 |
| CSDS | 353 | 274 | 80 | 164 | −65 | 105 | −16 | 161 | 110 | 145 |

89]. To further support our use of uniform swing in the Indian context, we provide an analysis of historical swings from 1989 (the first year election where the BJP/NDA was a serious contender) in S19 Fig in S1 Appendix. Note this analysis assumes the 2019 alliance-party composition, and is possible that under different formulations of these three alliances we may observe patterns of swing which are less compatible with the uniform hypothesis. Nevertheless, from our analysis it is clear that constituency swings appear to be symmetrically distributed around the national swing, suggesting that on average the uniform swing hypothesis holds in India.

Table 3 compares our estimates of the election result to state-of-the-art Indian election polling. In India, the most common opinion polls published on a relatively regular basis, before the beginning of the election period, are seats-projection polls. We obtain publicly available opinion polls for the 2019 election from Wikipedia (https://en.wikipedia.org/wiki/Opinion_polling_for_the_2019_Indian_general_election). For each poll reported on the Wikipedia page, we try to distinguish between the polling company and the news organization that commissioned the poll. Where no details on the polling company are available, we assume the news organization did the polling in-house. Pre-election polling stopped at around 40 days before voting begun; hence we compare our estimates at 5 weeks until the end of the election, with the last published poll of each of 10 polling houses that completed their last published poll at least a year before the last day of voting in 2019. Amongst the polling houses, we can identify three broad groups: 6 of the houses in question, namely 'CNX', 'Jan Ki Baat', 'VMR', 'News Nation', 'CVoter' and 'Zee 24 Taas' polled the campaign continuously, and published their last poll in the final week before voting begun—within 50 days to the end of the election. 3 pollsters, namely 'VDP Associates', 'Karvy' and 'India Tody', polled less regularly and published their last projections between 140 and 80 days to the end of the election. Finally, 'CSDS' and 'ABP News' released their last projection between a full year and 200 days to the end of the election. Fig 5 presents our seats projections and Table 3 describes how our error rates compare to other pollsters.

Firstly, it is immediately clear that our uniform-swing projections outperforms other pollsters: our poll has the lowest absolute error, 89 seats—together with the Jan Ki Baat poll; a gulf

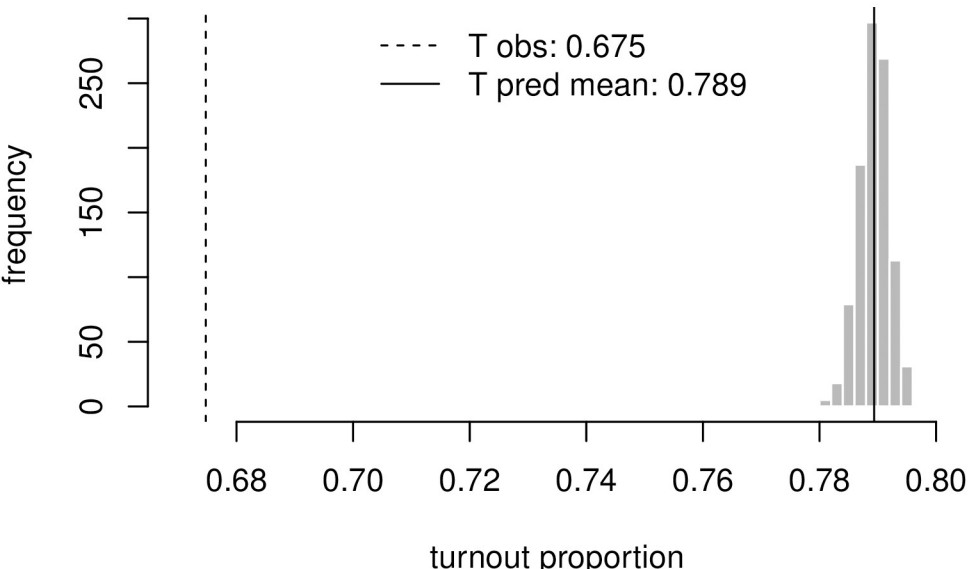

**Fig 5. Seats projections at 5 weeks till the end of voting.** *Pr*(majority) indicates the probability that a given party obtains an outright majority—272 seats or more—in the Lok Sabha.

of 54 stands between the top two performing polls and the next best. Importantly, our methodology vastly outperforms other pollsters on the NDA-UPA lead, missing the mark by a mere 8 seats, when compared to the next-best-performing Jan Ki Baat poll, which misses the mark by 67 seats. This is the most important predicted outcome because it typically directly determines government formation. At the alliance level, our method is the best performing on both NDA and UPA vote shares, with Jan Ki Baat coming in second for the NDA vote, and CNX for the UPA vote. Jan Ki Baat misses the NDA share by 44 seats, compared to our 27 seats error. The CNX forecast of the UPA seat share is off by 27 seats, compared to ours which is off by 18. Our seat forecast for the 'Other' category performs less well. As a final point, we also benchmark our uniform-swing predictions from the modeled data (regressed and post-stratified) against uniform swing projections based on the raw, un-modeled proportions form the online convenience samples. In every comparable prediction, our modeled forecasts outperform the raw un-modeled forecast. This further justifies the use of our complex modeling architecture to clearly add value to the non-representative sample at our disposal.

Fig 5 presents our seats projections incorporating their uncertainty intervals. The vertical red bar identifies the actual seat outcomes for the three categories. Finally, the uncertainty around the seats predictions is very narrow: the NDA forecast ranges 21 seats, between 321 and 342; the UPA forecast ranges 13 seats, between 73 and 86; and the 'Other' forecast ranges 15 seats, between 125 and 140. An indication of the over-confidence of the forecast is that even if our poll is the best performing, none of our intervals include the observed results.

## 7 Discussion

Forecasting the 2019 Lok Sabha election highlights the three challenges we face when implementing MrP modeling in challenging data contexts: a) producing an individual-level dataset

that allows for demographic profiles to be correlated with outcomes of interest; b) generating a stratification frame with precise population counts for all cells; and c) inplementing an algorithm that summarizes the relationships implied by the individual- level dataset and coherently generates 'out-of-sample' predictions.

It is increasingly challenging to generate survey samples that are representative of national populations—India is certainly not an exception. Online convenience samples that exploit novel technologies, namely social media and crowd-sourcing platforms are a promising solution. Our contribution is to suggest a mixed-mode online convenience sample that is cost-effective and reasonably powered for our 2019 India election forecast. A critical practical concern for area forecasts are the biases of these samples and how these might affect post-stratification predictions. We describe source-specific biases in the India case: Facebook users tend to be more upper caste, richer, and younger; Mechanical Turks are more diverse in terms of caste, more middle-class, overwhelmingly working age, and overwhelmingly from Tamil Nadu and Kerala; both groups are much more educated and richer than the vast majority of the Indian polity. As we point out below, the convenience samples are serviceable in spite of these biases. A question, unanswered here, is understanding how recruitment of these convenience samples can be optimized given the post-stratification frame and estimation goals.

A second challenge we address is the construction of a post-stratification frame. The Indian challenges are not dissimilar from what would be found in other national settings: a) random micro-data samples of the census are not available; b) public census tables are shallow and outdated; c) political variables are not included in the available joint distribution; d) nationally representative surveys are available, but they are heavily biased at the subnational level. We propose a novel methodology for constructing a post-stratification frame to obtain deep (i.e., highly disaggregated) and reliable estimates of the joint-distribution for the poststratification frame. Our methodology provides robust weights for these deep cells, by integrating multiple sources of population data.

These weights that we assign to each of the highly disaggregated, or deep, cells are estimated with uncertainty. A challenge, that we only tangentially address here, is how to account for uncertainty in the estimated weights. This has important implications because the precision of these weights, collectively, contribute significantly to uncertainty in our final aggregate-level voting predictions. The prediction intervals for our reported estimates are a case in point: as we pointed out, they are very narrow because our predictions are weighted averages across thousands of cells. By the simple logic of the central limit theorem, averaging over hundreds of cells leads to shrinking variance in the order of hundreds of times. Incorporating realistic uncertainty around the weights will reduce this shrinkage effect.

An important contribution here is simply illustrating the utility of existing probability machine methods for addressing estimation challenges associated with both the convenience sample-based individual level models and post-stratification. We suggest some innovative enhancements. We propose a beta-binomial model which enables accounting for prediction and sampling uncertainty via Monte Carlo simulations, leveraging assumptions about the predicted probabilities and their variance that are only mildly approximate. We further build on literature about the hyper-parameters of probability machines, and show that large values of `mtry` produce high-quality predictions. We further suggest `min.node.size` should be tuned to account for severe imbalance in a dichotomous outcome variable. Future work on probability machines should focus on developing a sampling theory for probability outputs.

We contribute to the three core elements of estimating or forecasting public opinion: the post-stratification frame, the convenience samples on which the individual-level modeling is based, and machine learning estimation methods. In virtually all national contexts, including India, the data requirements for the post-stratification frame and individual-level modeling

are imperfectly met. Technological advances, particularly with respect to the internet and computing, have significantly facilitated data generation and collection for both post-stratification frames and convenience samples. As a result we are able to collect much more diverse and voluminous data in both cases. Any one of these diverse data sources typically does not meet the very high standards of classic census data or national probability samples. We argue here though that highly precise public opinion estimates can be obtained by combining advanced machine learning modeling techniques with this increasingly diverse and voluminous, if imperfect, data. In this essay we illustrate this is the case by assembling imperfect convenience samples from Facebook and from AMTurk workers; identifying census micro-data and large-scale national survey data for the Indian population; and applying our novel machine learning estimation techniques. Our 2019 national forecasts of Lok-Sabha seat shares for the major competing alliances was spot-on, outperforming conventional public opinion forecasts of seat shares. This provides reassurances that this novel approach to both data collection and estimation is promising. We provide a road-map here for combining diverse data along with machine learning techniques in order to generate forecasts of public opinion and election seat outcomes. More generally, we believe these techniques are particularly valuable for regression and post-stratification modeling in contexts, like India, where data collection and generation are particularly challenging.

## Supporting information

**S1 Appendix.**
(PDF)

## Author Contributions

**Conceptualization:** Roberto Cerina, Raymond Duch.

**Data curation:** Roberto Cerina, Raymond Duch.

**Methodology:** Roberto Cerina.

**Resources:** Raymond Duch.

**Software:** Roberto Cerina.

**Supervision:** Raymond Duch.

**Writing – original draft:** Roberto Cerina, Raymond Duch.

**Writing – review & editing:** Raymond Duch.

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
