## [Decision Letter · Decision Letter 0]

25 May 2021

PONE-D-21-12678

Polling India via Regression and Post-Stratification of Non-Probability Online Samples

PLOS ONE

Dear Dr. Cerina,

Thank you for submitting your manuscript to PLOS ONE. After careful consideration, we feel that it has merit but does not fully meet PLOS ONE’s publication criteria as it currently stands. Therefore, we invite you to submit a revised version of the manuscript that addresses the points raised during the review process.

I have not heard from two very knowledgeable reviewers and they both have recommended revise and resubmit. I have read the draft and given my interest in this topic, I found the paper a good fit for PLOS ONE. However, both reviewers have listed several suggestions that I think you should address when you resubmit the report. These include the writing of the paper, being very clear about what your model can predict, or the limitations.

In addition to the comments from the two reviewers, I was interested in knowing how will your model predict -- state elections in Tamil Nadu, Kerala, Rajasthan -- these are states that are known to change the ruling party every time it goes for elections.

I look forward to seeing the revised version soon.

Kind regards,

Nishith Prakash, Ph.D.

Academic Editor

PLOS ONE

Additional Editor Comments:

Dear Dr. Cerina

I have not heard from two very knowledgeable reviewers and they both have recommended revise and resubmit. I have read the draft and given my interest in this topic, I found the paper a good fit for PLOS ONE. However, both reviewers have listed several suggestions that I think you should address when you resubmit the report. These include the writing of the paper, being very clear about what your model can predict, or the limitations.

In addition to the comments from the two reviewers, I was interested in knowing how will your model predict -- state elections in Tamil Nadu, Kerala, Rajasthan -- these are states that are known to change the ruling party every time it goes for elections.

I look forward to seeing the revised version soon.

Best,

Nishith

Journal Requirements:

Reviewers' comments:

Reviewer's Responses to Questions

**Comments to the Author**

1. Is the manuscript technically sound, and do the data support the conclusions?

Reviewer #1: Yes

Reviewer #2: Partly

2. Has the statistical analysis been performed appropriately and rigorously? 

Reviewer #1: Yes

Reviewer #2: Yes

3. Have the authors made all data underlying the findings in their manuscript fully available?

Reviewer #1: Yes

Reviewer #2: No

4. Is the manuscript presented in an intelligible fashion and written in standard English?

Reviewer #1: Yes

Reviewer #2: Yes

5. Review Comments to the Author

Reviewer #1: This paper proposes and implements a method for using convenience online polling data (Facebook and Amazon’s Mechanical Turk) to predict election outcomes in India. A simplified version of the steps of that process are the following: 1) use Indian census data to estimate the fraction of individuals in the population with particular combinations of characteristics; 2) generate a simulated micro-data sample based on that; 3) use the Indian National Election Survey and Indian Human Development Survey to impute characteristics for observations in that simulated microdata, such as past turnout; 4) calculate weights for the observations in the microdata so that statistics generated from that data will match aggregate statistics such as turnout; 5) use online polling data (from Facebook and Amazon’s Mechanical Turk) from 2019 to model the relationship between individual-level characteristics and voting choices; 6) use data from the 2014 INES to model the relationship between individual-level characteristics and turnout; and 7) take that estimated relationship to the microdata to estimate turnout and voting decisions. The authors published predictions based on this methodology prior to the 2019 elections and these were more successful than public polls of the results. This suggests that this process for combining data sources can be used to make successful all-India election predictions.

Main comments:

1. This paper addresses a difficult problem for which there are no perfect solutions – it is challenging to predict election outcomes in contexts where true random sampling is not possible. However, there are a number of challenges for the voter turnout modeling in the paper. The 2019 online polling data on intended turnout cannot be used to create a good turnout model since 97% of the sample reports that they will turn out to vote. The authors instead use the 2014 INES data, but even that is not very helpful for building a turnout model since nearly everyone reports having turned out to vote in that survey. As a result, the turnout predictions are poor: for example, looking at the state-level predictions of turnout in the upper right figure of figure S17, the correlation between predicted and actual turnout is low and would be basically zero with one state excluded.

I do not think that the authors need to change this aspect of the paper in a revision – one of the strengths of the paper is that the authors made public predictions prior to the elections based on this model and then validated them against the actual election results. Instead, I think the authors should revise the paper to describe these weaknesses more clearly in the paper. As I see it, there are two points to highlight. First, they are using 2014 data to model 2019 turnout. As a result, their method would struggle in elections with differential turnout patterns across cycles. The authors should be clear about this point.

Second, it is more accurate to describe their method as based almost entirely on estimating voter preferences rather than about turnout; they just don’t have data that would allow them to accurately predict turnout. That is fine as a limitation of this particular method, but it should be made clear to readers that they are really just trying to peg overall turnout to 2014 and then model voter preferences as a function of observable characteristics.

2. The authors are transparent in noting many of the limitations of their approach. I thought that there were a number of places where the authors need to more heavily caveat some weakness of their approach and how these weaknesses could cause their method to yield misleading estimates in other election cycles. This will also help readers who are interested in using similar methodologies to build future models of Indian elections.

a. The key reason that the method of authors was better at predicting election outcomes was that they were more optimistic about BJP performance and pessimistic about UPA performance than the public pollsters. It was not clear to me whether this is because of particularly strength of their methodology or because online samples happen to have unobservable characteristics that predispose them towards BJP (and against UPA) beyond that would be expected by observable characteristics. It seems like that may be true (page 21), and as a result, it may not be that the method is generally strong, but that in an election where UPA underperforms polls/BJP overperforms, this bias happened to push them in the right direction.

b. In section 4.1, the “across the board correction” should be more accurately described as randomly selecting around a third of the sample to change the value of their reported turnout to zero. This is not a “correction” -- even under the very strong (and probably incorrect) assumption that misreporting past voter status is unrelated to characteristics of the individual, this is just introducing a bunch of noise into the predictions. The authors should be clearer about what this process is doing and the assumptions required for this to be asymptotically valid.

c. On pg 20, the paper describes it as an “oddity” that Christians are over-represented among Mechanical Turks as well as some other issues with that sample (e.g. more likely to support NDA/UPA). In all cases, it seemed clear that this was because Tamil Nadu and Kerala make up over 80% of the Mechanical Turk sample. For example, Kerala has the highest density of Christians in India. This could be mentioned there as the likely explanation. In general, given the geographical heterogeneity of India, it seemed like the Facebook data is more promising for all-India predictions in future election cycles.

d. On pg 13, the authors refer to deeper selection issues relating to the NES but do not describe them: I would appreciate it if the authors made these more explicit at that point or at least referred to points later in the paper that discuss this and how these may effect their overall results.

Minor comments:

1. I wasn’t able to access the data or find it on the Harvard dataverse that they report uploading it to, so they should update the links (or perhaps make it publicly searchable)

2. In section 3.2, the process for raking the NES data to the known 2014 election results at the state level was not clear – what variables were used when generating the raking weights? This should be described more carefully.

3. Rahul Gandhi’s name is misspelled on page 5.

4. On page 17, I did not understand why there were more voting intentions than there were users – is this a single IP address completing the survey multiple times? Please explain in more detail.

5. In figure S5, there should be a legend added to explain the color coding used there.

Reviewer #2: Summary

The paper proposes a method of data collection and analysis to project electoral outcome during campaign in the Indian context. The authors conduct repeated online polling of a non-representative sample of voters and use various census and survey datasets to project voting behavior for the entire population to estimate the aggregate turnout, vote shares and seat shares of the main coalitions for the 2019 general election in India. The method produces very good predictions for all the three measures, and therefore, may have the potential to be used more widely to get better early predictions of election outcomes. Below I describe my comments highlighting various issues with the paper.

Comments

1. The first major issue with the paper is its writing. The authors have not managed to articulate their work very well. Consequently, I was left confused in many segments and had to make educated guesses about what they were doing. The LHS of equation (1), for example, is defined simply as “vote choice” of the Indian voters. From discussions several pages later it appears that it is a vector of vote shares of three coalitions. It is not mentioned what is “X” or the index “i” in equation (1) that the authors are summing over. The same equation appears again later in the paper, where some of the expressions are explained. Similarly, it is not well-explained what is being depicted in Figure 1. Specifically, what do the points in the graph represent? The paper needs significant rewriting to make the details clear.

2. The main contribution of the work seems to be predicting the aggregate vote shares of the three coalitions (NDA, UPA and Others) quite well. The prediction is a combination of two exercises – continuous and real time opinion polls in two online platforms and projecting the reported behavior of this non-representative sample on the population. Since both exercises are “new” relative to the existing opinion polls in India (more on this in the next point), it would be good to know the relative importance of each exercise in ensuring a good prediction. Specifically, what would the vote share predictions be if the opinion polls are taken as they are, without adjusting for the non-representativeness of the underlying sample? This would tell us how much of additional improvement in prediction the authors achieve by employing the statistical technique.

3. In relation to the above point a discussion of a particular contextual detail may be important. There was a major incident that happened before Lok Sabha election in 2019 that became politically highly salient. On February 26, 2019, there was a terror attack on an army convoy in Kashmir, which escalated into a major tension between India and Pakistan, from which the incumbent government seemed to have electorally gained in the 2019 elections. The incident happened around 12 weeks prior to the start of the elections, exactly around the time Figure 3 depicts the vote shares. As evident from the Figure, the NDA’s vote share rises sharply around that time, before stabilizing around week 9. This shift in behavior would not be picked up in the other opinion polls as most of them were conducted before the incident, as Table 3 shows. In that sense, (repeated) online opinion polls have the advantage of picking up shocks to voter preferences that occur closer to elections that standard survey based opinion polls may not be able to capture. The authors should highlight this point. Moreover, if the shift in preference in favor of NDA was uniform across population groups, then the additional advantage of using the authors’ statistical method may not be that high in this specific application. It is therefore important for them to demonstrate the marginal contribution of the method.

4. The method to convert the vote shares to seat shares is rather crude, and in general, it would not work. My sense is that it worked well in this specific case because the shift in vote in favor of NDA in 2019 relative to 2014 was mostly uniform across constituencies, thanks to the general positive attitude in favor of the incumbent Prime Minister, who was the main face of NDA. I suggest the authors move the entire discussion on seat share to the appendix, unless they are able to propose a significantly better method.

5. The authors use NES 2014 to compute turnout probabilities. Given the really high reported turnout even in NES 2014, it seems there are other options. For example, the authors can use constituency level true turnout figures and regress them on constituency level demographic and other variables (available from SHRUG data: http://www.devdatalab.org/shrug) to get at how population characteristics map to turnout behavior.

6. PLOS authors have the option to publish the peer review history of their article (what does this mean?). If published, this will include your full peer review and any attached files.

Reviewer #1: No

Reviewer #2: No

---

## [Author Response · Author response to Decision Letter 0]

7 Jul 2021

We have uploaded a separate document with responses to the reviewers' comments.

---

## [Decision Letter · Decision Letter 1]

20 Oct 2021

PONE-D-21-12678R1

Polling India via Regression and Post-Stratification of Non-Probability Online Samples

PLOS ONE

Dear Dr. Cerina,

Thank you for submitting your manuscript to PLOS ONE. Both reviewers are happy with the revised version, but want you to address few additional comments. I think the comment can be easily taken care of.

Please send the revised draft after addressing both R1 and R2 carefully. I look forward to reading the revised draft soon.

We look forward to receiving your revised manuscript.

Kind regards,

Nishith Prakash, Ph.D.

Academic Editor

PLOS ONE

Journal Requirements:

Reviewers' comments:

Reviewer's Responses to Questions

**Comments to the Author**

1. If the authors have adequately addressed your comments raised in a previous round of review and you feel that this manuscript is now acceptable for publication, you may indicate that here to bypass the “Comments to the Author” section, enter your conflict of interest statement in the “Confidential to Editor” section, and submit your "Accept" recommendation.

Reviewer #1: All comments have been addressed

Reviewer #2: (No Response)

2. Is the manuscript technically sound, and do the data support the conclusions?

Reviewer #1: Yes

Reviewer #2: Yes

3. Has the statistical analysis been performed appropriately and rigorously?

Reviewer #1: Yes

Reviewer #2: Yes

4. Have the authors made all data underlying the findings in their manuscript fully available?

Reviewer #1: Yes

Reviewer #2: No

5. Is the manuscript presented in an intelligible fashion and written in standard English?

Reviewer #1: Yes

Reviewer #2: Yes

6. Review Comments to the Author

Reviewer #1: The authors have done a good job responding to all of my comments. After thoroughly re-reading the paper and the reply letter, I am satisfied with all of the revisions and have no major additional comments. I had a few typographical and formatting comments, but otherwise am very happy with the revised version of the paper.

Minor Comments:

- Footnote 12: “social trus “ should be “social trust.” The footnote should also come after the period rather than before.

- Footnote 22: this footnote should be placed after the period rather than before.

- Page 21: please include a reference to the figure numbers for the sample v population plots in the appendix so it is easier for the reader to find them.

- Page 29: “advice fore the” should be “advice for the”

- Page 33: “attributable to the quality the online pre-election polling” should be “to the quality of the online pre-election polling”. The sentence “put plainly, to what extent is our prediction more accurate as a result of modeling conditional on turnout ? and is the turnout modeling architecture justified ?” should be “put plainly, to what extent is our prediction more accurate as a result of modeling conditional on turnout, and to what extent is the turnout modeling architecture justified?”

- Fig S21: “convenicence” should be “convenience”

Reviewer #2: The authors have addressed some of the issued raised in my first review. The detailed comments are in attachment.

7. PLOS authors have the option to publish the peer review history of their article (what does this mean?). If published, this will include your full peer review and any attached files.

Reviewer #1: No

Reviewer #2: No

---

## [Author Response · Author response to Decision Letter 1]

1 Nov 2021

A rebuttal letter has been attached to the submission.

---

## [Editor Report · Decision Letter 2]

3 Nov 2021

Polling India via Regression and Post-Stratification of Non-Probability Online Samples

PONE-D-21-12678R2

Dear Dr. Cerina,

We’re pleased to inform you that your manuscript has been judged scientifically suitable for publication and will be formally accepted for publication once it meets all outstanding technical requirements.

Kind regards,

Nishith Prakash, Ph.D.

Academic Editor

PLOS ONE
---

## [Editor Report · Acceptance letter]

8 Nov 2021

PONE-D-21-12678R2 

Polling India via Regression and Post-Stratification of Non-Probability Online Samples 

Dear Dr. Cerina:

I'm pleased to inform you that your manuscript has been deemed suitable for publication in PLOS ONE. Congratulations! Your manuscript is now with our production department. 

Kind regards, 

on behalf of

Dr. Nishith Prakash 

Academic Editor

PLOS ONE